# Community effects in regulation of translation

Paul M Macdonald[1,2]*, Matt Kanke[1,2†], Andrew Kenny[1,2]

[1]Department of Molecular Biosciences, The University of Texas at Austin, Austin, United States; [2]Institute for Cellular and Molecular Biology, The University of Texas at Austin, Austin, United States

**Abstract** Certain forms of translational regulation, and translation itself, rely on long-range interactions between proteins bound to the different ends of mRNAs. A widespread assumption is that such interactions occur only in *cis*, between the two ends of a single transcript. However, certain translational regulatory defects of the *Drosophila oskar (osk)* mRNA can be rescued in *trans*. We proposed that inter-transcript interactions, promoted by assembly of the mRNAs in particles, allow regulatory elements to act in *trans*. Here we confirm predictions of that model and show that disruption of PTB-dependent particle assembly inhibits rescue in *trans*. Communication between transcripts is not limited to different *osk* mRNAs, as regulation imposed by *cis*-acting elements embedded in the *osk* mRNA spreads to *gurken* mRNA. We conclude that community effects exist in translational regulation.

*For correspondence: pmac@utexas.edu

Present address: †Department of Genetics, University of North Carolina at Chapel Hill, Chapel Hill, United States

Competing interests: The authors declare that no competing interests exist.

## Introduction

Translation happens when a ribosome assembles productively on an mRNA. A number of molecular interactions influence the proper convergence of these parts, often involving RNA elements or factors at the opposite ends of the mRNA (*Jackson et al., 2010*). For example, Poly(A) binding protein (PABP) binds to the poly(A) tail at the 3' end of an mRNA (*Mangus et al., 2003*) and enhances translation through its interaction with eIF4G, a protein associated with the 5' cap of the mRNA (*Tarun et al., 1997*; *Sonenberg and Dever, 2003*). Similarly, many repressors of translation bind to elements in the 3' untranslated regions (3' UTRs) of mRNAs and interfere with the normal action of initiation factors at the 5' end of the mRNA (*Jackson et al., 2010*; *Gebauer and Hentze, 2004*). Such long-range protein-protein interactions, spanning the full length of the mRNA, are generally assumed to occur only in *cis*: PABP bound to the tail of one molecule of mRNA is expected to bind only the eIF4G associated with the other end of the same mRNA. In principle, however, inter-transcript interactions are also possible, and their likelihood depends on the concentration of the mRNAs. When mRNAs are dilute, intra-transcript interactions will predominate, but as the local density of mRNAs increases, inter-transcript interactions become more likely.

Previously, we described an example involving the *Drosophila oskar (osk)* mRNA, in which control elements on one transcript influenced translation of another transcript (*Reveal et al., 2010*). *osk* mRNA is expressed during oogenesis, and is subject to multiple forms of post-transcriptional control. Following transcription in the nurse cells of each egg chamber, *osk* mRNA is transported through cytoplasmic connections to the transcriptionally-silent oocyte. Eventually, as oogenesis proceeds, *osk* mRNA localizes to the posterior pole of the oocyte, and only then does Osk protein begin to accumulate. Translation of *osk* mRNA is highly regulated: repression prevents premature translation from unlocalized mRNA, and activation turns on translation of the localized mRNA (*Kim-Ha et al., 1995*; *Markussen et al., 1995*; *Rongo et al., 1995*). Both types of regulation rely on binding sites for Bruno (Bru) (*Reveal et al., 2010*; *Kim-Ha et al., 1995*; *Webster et al., 1997*;

**eLife digest** Genes encode the instructions needed to make proteins and other molecules. To make a protein, the DNA within a gene is copied to produce molecules of messenger ribonucleic acid (mRNA) that are then used as templates to build proteins via a process called translation. This process – which involves protein machines called ribosomes binding to the start of the mRNA – is tightly regulated to control the amounts of particular proteins in cells. For example, in fruit fly ovaries, a protein called Bruno both represses and activates the translation of a gene known as *oskar*. To achieve this, Bruno binds to regions near the end of the *oskar* RNA known as Bruno response elements.

It is not clear how Bruno acts to control translation. However, because ribosomes begin translation near the start of the mRNA, while Bruno is bound to regions near the end of the mRNA, there must be long-range interactions between the two ends of the mRNA. It is generally assumed that such long-range interactions only occur between proteins that are bound to the same mRNA molecule. However, in 2010, researchers observed that Bruno response elements within one *oskar* mRNA could influence the translation of other *oskar* mRNAs. This is known as "regulation in *trans*". Here, Macdonald et al. – including some of the researchers from the earlier work – investigated this observation in more detail in fruit flies.

In cells, multiple mRNA molecules and their associated proteins can assemble into particles. Macdonald et al. proposed that the close proximity of many mRNA molecules in these particles could allow *trans* regulation to take place. Indeed, the experiments found that blocking the assembly of *oskar* mRNA into particles inhibited *trans* regulation as expected. Macdonald et al. also asked if *trans* regulation can occur between mRNAs that encode different proteins. The experiments show that *oskar* mRNA could block the translation of an mRNA produced by the *gurken* gene, even when *oskar* mRNA was not being translated. More work is needed to find out how widely *trans* regulation is used to control translation.

*Reveal et al., 2011*). These sites, BREs and others, reside in two clusters - the AB and C regions - near the opposite ends of the 3' UTR (*Figure 1A*). All the BREs contribute to translational repression (*Kim-Ha et al., 1995*), and the C region BREs also play a role in translational activation (*Reveal et al., 2010*). Strikingly, defects in either repression (from mutation of all BREs in *osk ABC BRE⁻*) or activation (from mutation of the C BREs in *osk C BRE⁻*) can be rescued by the presence of another *osk* mRNA, one that itself cannot make Osk protein but has all regulatory elements intact (*Reveal et al., 2010*).

To explain rescue in *trans,* we proposed that long-range protein-protein interactions bridging the two ends of an mRNA could also bridge two different mRNAs, to enable 'regulation in *trans*'. A proposed mechanism for translational repression by Bru relies on such a long-range interaction. Bru binds Cup, a protein that competes with eIF4G for binding to eIF4E, the 5' cap-binding protein. Cup engagement of eIF4E at the expense of the eIF4E/eIF4G interaction blocks translation initiation (*Nakamura et al., 2004*). In our model, the Bru/Cup complex bound to the rescuing mRNA would bridge to eIF4E bound to the cap of the mRNA unable to bind Bru, thereby conferring translational repression. We hypothesized that assembly of *osk* mRNAs into ribonucleoprotein particles (RNPs) places the mRNAs in close proximity, thus increasing their concentration and facilitating inter-transcript interactions to allow 'regulation in *trans*' (*Reveal et al., 2010*).

If this hypothesis is correct, several features of 'regulation in *trans*' are expected. First, the rescuing (donor) mRNA should have the regulatory element that the rescued (recipient) mRNA lacks. Second, rescue in *trans* should not be Bru-specific, but should extend to other forms of regulation that also rely on long-range protein-protein interactions. Finally, rescue in *trans* would depend on the *cis*-acting *osk* mRNA elements and *trans*-acting proteins that mediate RNP assembly.

Several participants in *osk* RNP assembly are known - Polypyrimidine tract binding protein (PTB), Bru and an RNA dimerization motif - and all are candidates to support rescue in *trans*. PTB binds *osk* mRNA to form RNPs in vitro (*Besse et al., 2009*). PTB also mediates association of *osk* mRNAs in vivo, as shown by the effects of loss of PTB on hitchhiking (*Besse et al., 2009*), the phenomenon in

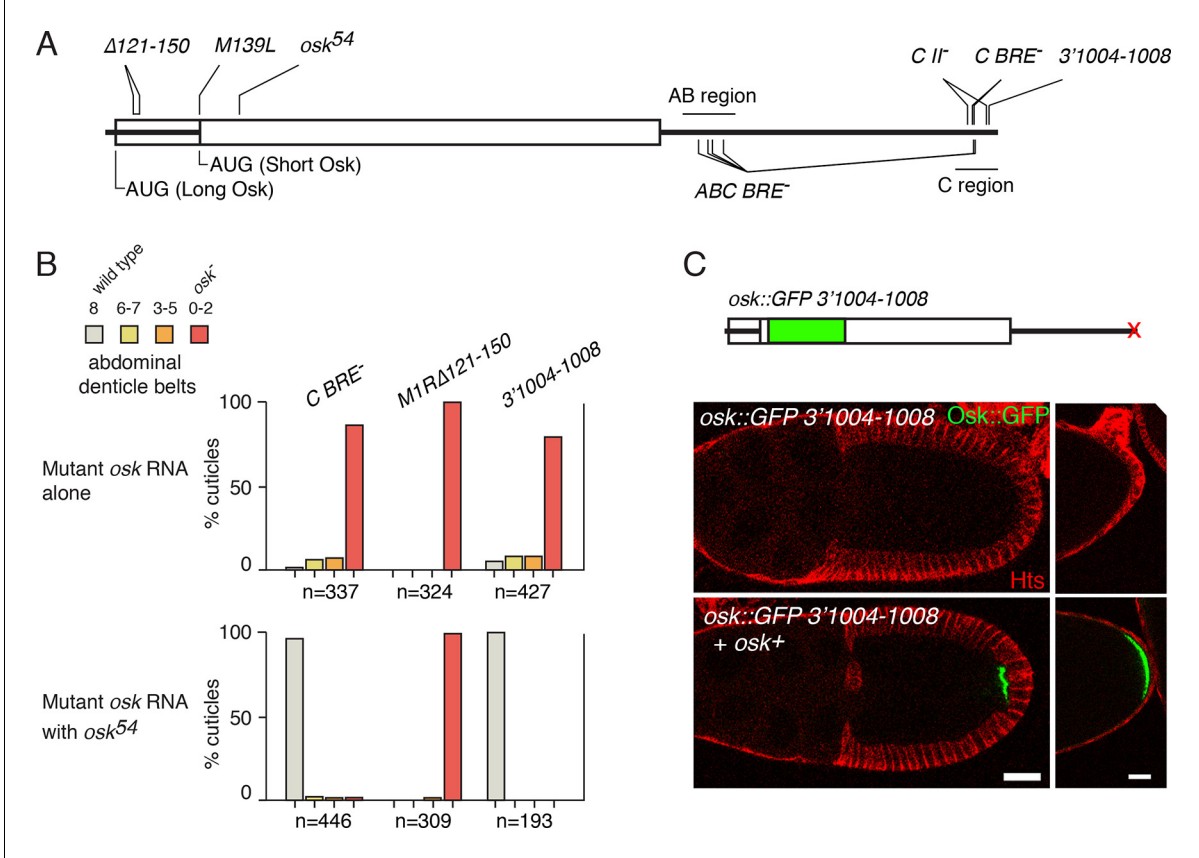

**Figure 1.** Rescue of *osk* expression in *trans* is not limited to Bru-dependent regulation. (**A**) Diagram of the *osk* mRNA showing sites of mutations discussed in the text. The UTRs are shown as thick lines and the coding region as a rectangle. Because two different start codons are used, portions of the 5' region can be both UTR and coding region. Not all the mutations at the 3' end of the mRNA are shown. (**B**) Embryonic patterning assays to monitor *osk* activity. For the upper panel, the only source of *osk* mRNA was a genomic *osk* transgene with the indicated mutation. For the lower panel, *osk54* mRNA was also present. Embryos from these mothers were were scored for cuticular patterning defects. Wild-type embryos have eight abdominal segments. Lower *osk* activity results in fewer abdominal segments. n values are the number of embryos scored. (**C**) Rescue in *trans* of the Osk expression defect caused by mutation 3'1004–1008. At top is a diagram of the mRNA from the genomic *osk* transgene whose activity was monitored. All sequences are from *osk*, except for the inserted GFP. Below are images of stage 10 egg chambers (left) and the posterior ends of late stage egg chambers (right). Scale bars, 25 μm.

which localization-competent *osk* mRNAs can assist other mRNAs in moving to the posterior pole of the oocyte (*Hachet and Ephrussi, 2004*). Similarly, Bru mediates in vitro oligomerization of RNAs with an unusually high density of Bru binding sites and could also oligomerize *osk* mRNA in vivo (*Chekulaeva et al., 2006*). Finally, *osk* mRNA undergoes direct RNA dimerization, relying on a dimerization motif in the 3' UTR (*Jambor et al., 2011*). Elimination of one or more types of *osk* RNP assembly would be expected to block rescue in *trans* if our hypothesis is correct.

Here, we show that all these expectations for the 'regulation in *trans*' model are met. Disruption of individual contributions to RNP assembly reveals that PTB has a critical role in rescue in *trans*, with direct RNA dimerization being much less important. Furthermore, we provide evidence that regulation imposed by embedded *cis*-acting elements in *osk* mRNA spreads to the another mRNA bound by PTB, the *gurken (grk)* mRNA, thus revealing a community effect in translation.

## Results

### Many, but not all, *osk* regulatory mutants can be rescued in *trans*

Previously, we found that translational regulatory defects associated with mutation of Bru binding sites were rescued in *trans*. To determine if rescue in *trans* is a property unique to Bru-dependent regulation, or more general, we have now identified and tested additional regulatory mutants.

The *osk* gene produces two Osk isoforms, Long Osk and Short Osk, from two translation start codons (*Figure 1A*) (*Markussen et al., 1995*). The *oskM1RΔ121–150* mutant lacks the start codon for Long Osk and has a 30 nt deletion in the 5' UTR for Short Osk (*Figure 1A*). This mutant is largely defective in Short Osk protein production, in addition to making no Long Osk because of the M1R mutation (*Kanke and Macdonald, 2015*). Although coexpression with *osk54* (a protein null mutant with a short DNA insertion, introducing a stop codon at the end of the first exon) (*Kim-Ha et al., 1991*) strongly rescues the activation defect of *osk C BRE⁻* (*Figure 1B*), no significant rescue of the *oskM1RΔ121–150* activation defect was found (*Figure 1B*). Therefore, the *oskM1RΔ121–150* mutant is not sensitive to rescue in *trans*.

Mutations positioned close to the 3' end of the *osk* mRNA also disrupt Osk expression. Mutants *osk3'977–981* and *osk3'984–988* (numbering indicates position in the *osk* 3' UTR) have almost no *osk* patterning activity and are strongly defective in Osk protein production (*Ryu and Macdonald, 2015*). Mutant *osk3'1004–1008*, in which an A-rich sequence is disrupted (*Kanke et al., 2015*) (*Figure 1A*), also had strongly reduced *osk* patterning activity (*Figure 1B*). Although these mutations are positioned close to Bru-binding sites in the *osk* 3' UTR C region, they have little or no effect on Bru binding (*Kim-Ha et al., 1995*; *Ryu and Macdonald, 2015*). Instead, the 977–980 and 984–988 mutations disrupt binding of BSF protein (*Ryu and Macdonald, 2015*), while the sequence affected by mutation 1004–1008 is a binding site for Poly(A) binding protein (PABP) (*Vazquez-Pianzola et al., 2011*). The patterning defects of all three of these mutants were strongly rescued by coexpression of *osk54* mRNA (*Figure 1B*, [*Ryu and Macdonald, 2015*]). Similarly, a more direct assay of Osk protein expression, using an *osk::GFP* transgene bearing the *osk3'1004–1008* mutation, showed rescue in *trans* (*Figure 1C*).

From analysis of these four additional regulatory mutants, we find a correlation (albeit with few examples) between the type of regulatory defect and its ability to be rescued in *trans*: regulatory elements likely to engage in long-range interactions (i.e. those in the 3' UTR) are sensitive to rescue in *trans*, while a regulatory element positioned near the site of translational initiation and thus less likely to participate in a long-range interaction is not. Notably, by the model of 'regulation in *trans*' only the former type of regulatory element should be sensitive to rescue in *trans*, consistent with the results.

### Many mutant *osk* mRNAs can act as donors for rescue in *trans*

Thus far, *osk54* has been used as the donor for rescue in *trans* of *osk* mRNA regulatory defects. To determine if the ability to rescue is typical, or is an unusual property of *osk54*, additional *osk* mutants unable to provide Osk patterning activity were tested. In *oskM139L*, the translation start codon for the Short Osk protein isoform is mutated. Because Short Osk is the isoform with embryonic patterning activity, *oskM139L* lacks that activity (*Vanzo and Ephrussi, 2002*). Like *osk54*, *oskM139L* strongly rescued the translational activation defect of *osk C BRE⁻* and restored normal embryonic patterning (*Figure 2A*). Likewise, *oskM1RΔ121–150*, the mutant whose regulatory defect is not rescued in *trans*, was an effective donor for rescue of the *osk C BRE⁻* mutant in *trans* (*Figure 2A*). Rescue was also assayed by examination of Osk protein production, using the *osk::GFP C BRE⁻* mRNA as the recipient and monitoring GFP fluorescence (this allows the mRNA producing the protein to be unambiguously identified, as the donor mRNAs do not include GFP). Both *oskM139L* and *oskM1RΔ121–150* restored Osk::GFP protein expression to wild-type levels, just as with the *osk54* donor (*Figure 2C*).

These results, and others (below), show that a variety of *osk* mutants can act as donors for rescue in *trans*; this property is not unique to *osk54*. Furthermore, rescue by *oskM1RΔ121–150* is consistent with 'regulation in *trans*', as the *oskM1RΔ121–150* mRNA includes the regulatory element that the *osk::GFP C BRE⁻* mRNA lacks.

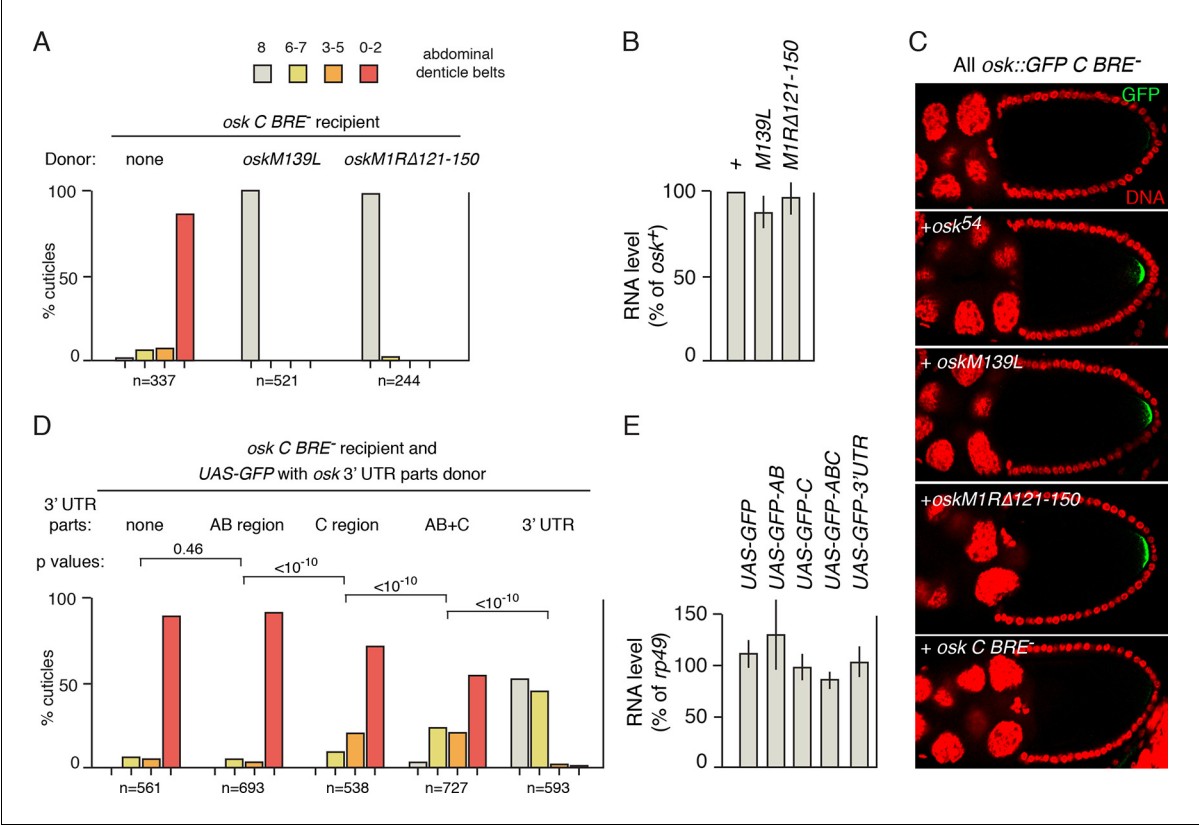

**Figure 2.** Identification of sequences required in donor mRNAs for rescue in *trans*. (A) Embryonic patterning assays to monitor rescue of the translational activation defect of *osk C BRE*⁻. The *osk C BRE*⁻ mRNA was expressed as the only *osk* mRNA, or in combination with donor *osk* mRNAs as indicated. (B) Levels of transgenic *osk* mRNAs. Values are relative to an *osk*⁺ transgene that fully rescues an *osk* mutant. Error bars indicate standard deviations. The *osk C BRE*⁻ mRNA used in panel C is not included but has similar abundance (*Reveal et al., 2010*). (C) Detection of GFP signal from *osk::GFP C BRE*⁻, expressed as the only *osk* mRNA or in combination with the donor *osk* mRNAs indicated. (D) Embryonic patterning assays to monitor rescue of the translational activation defect of *osk C BRE*⁻. The *osk C BRE*⁻ mRNA was expressed as the only *osk* mRNA (in the *oskA87/osk0* background), together with a version of *UAS-GFP* expressed under GAL4 transcriptional control. Each *UAS-GFP* transgene contains *osk* 3' UTR sequences as indicated (positions of the AB and C regions are shown in *Figure 1A*). p values from student's t tests were determined as described in 'Experimental Procedures', with comparison sets highlighting the incremental increases in rescuing activity resulting from the presence of the C region together with additional parts of the 3' UTR. Values for effect size and power were: AB vs C, 0.444 and 1.0; C vs AB+C, 0.448 and 1.0; AB+C vs *osk* 3' UTR, 2.21 and 1.0. Additional student's t tests for comparison of GFP alone to GFP plus the C, AB+C and 3' UTR parts all had p values of <10⁻¹⁰. (E) Levels of mRNAs from D. Error bars indicate standard deviations.

## Missing regulatory elements are required, but not sufficient, for rescue in *trans*

The model for 'regulation in *trans*' predicts that the regulatory element missing from the recipient mRNA must be provided by the donor. This idea can be tested with *osk::GFP C BRE*⁻ as the recipient, and *osk C BRE*⁻ as the donor, as both carry the same regulatory mutation. Although the activation defect of *osk::GFP C BRE*⁻ was strongly rescued by *osk54* (and by other donors with intact C region BREs; *Figure 2C*, below), little or no GFP signal was detected with *osk C BRE*⁻ as the donor (*Figure 2C*).

Having shown that rescue of a regulatory defect in the recipient mRNA requires the cognate, wild type regulatory element in the donor mRNA, we next asked if the presence of that element was sufficient for rescue, or if other sequences (e.g. those that might contribute to RNP assembly) were also required. Portions of the *osk* mRNA 3' UTR were added to a *UAS-GFP* transgene, and expressed using the UAS/GAL4 system (*Brand and Perrimon, 1993*). The *osk* 3' UTR AB and C regions contain the 5' and 3' clusters of Bru binding sites, respectively (*Figure 1A*). These regions were tested alone, or in combination. Also tested was the entire *osk* 3' UTR.

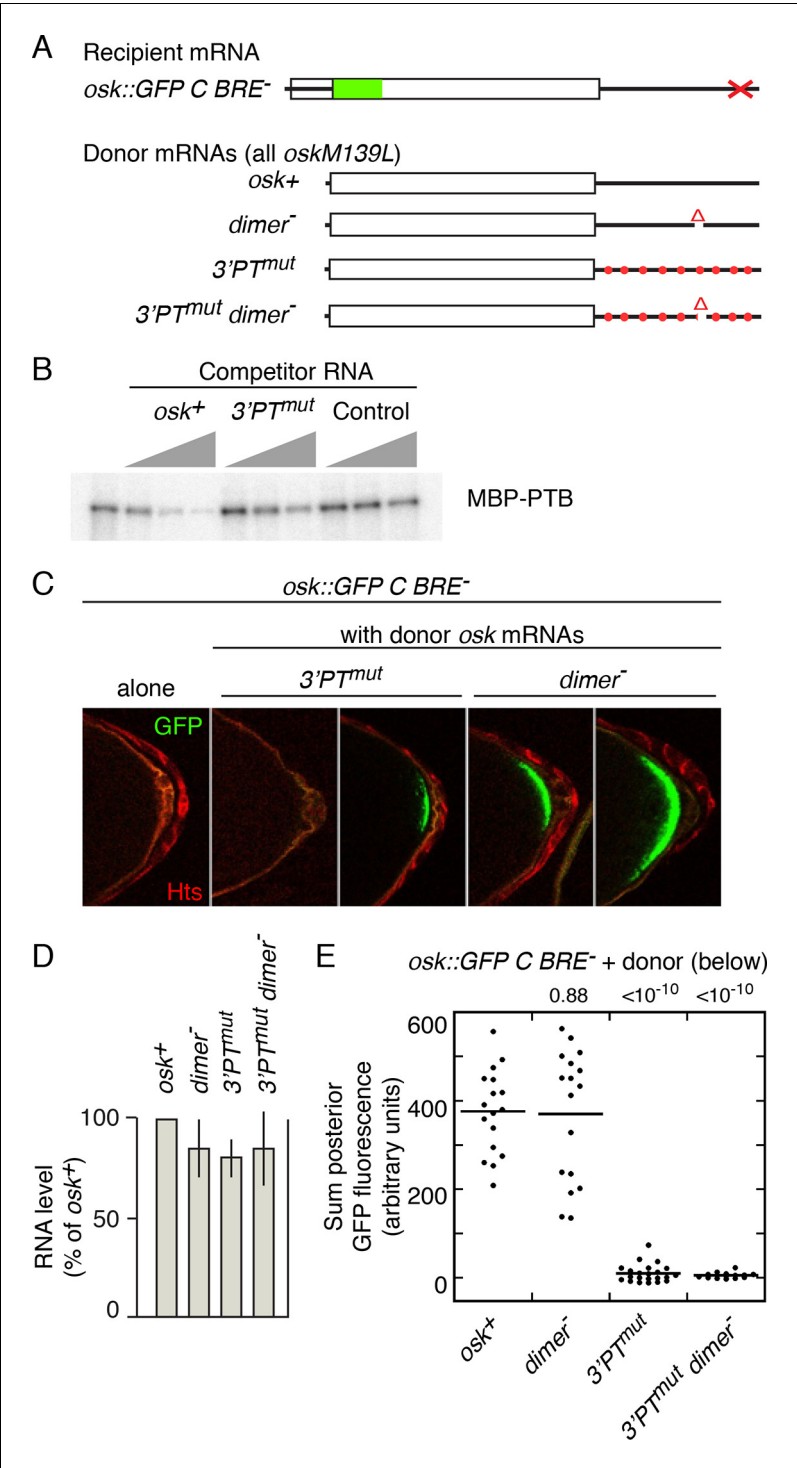

**Figure 3.** Mutation of pyrimidine tracts in the *osk* mRNA 3′ UTR disrupts rescue in *trans*. (**A**) Diagram of *osk* mRNAs used in the assays. The *dimer⁻* mRNA has a deletion of positions 665–685 of the 3′ UTR. The *3′PT^mut* mRNA has multiple mutations (**Figure 3—figure supplement 1**), indicated figuratively by the red dots. Each donor *osk* transgene for rescue has the M139L mutation to prevent translation of the Short Osk isoform. Only Short Osk has patterning activity (**Vanzo and Ephrussi, 2002**), and we eliminated this isoform to address the possibility that some mutations might cause inappropriate Osk expression and dominant maternal-effect lethality from excess *osk* patterning activity. (**B**) PT mutations disrupt PTB binding. UV crosslinking assay of MBP-PTB binding to the *osk* 3′ UTR. The left lane shows binding to the radiolabeled *osk* 3′ UTR RNA in the absence of competitor RNA. For the remaining lanes, increasing amounts (3, 10, 30x molar excess) of unlableled competitor

*Figure 3 continued on next page*

*Figure 3 continued*

RNAs were included in the binding assays. The competitor RNAs were the wild type *osk* 3' UTR (*osk*+), the *osk* 3' UTR with the PT mutations included in the *osk 3'PT^mut* transgene (*3'PT^mut*), or a nonspecific control RNA corresponding to a portion of the *bicoid* mRNA 3' UTR (segment 3R: 6756590–6756463, r6.08) with no strong predicted PTB binding sites (Control). (**C**) Examples of oocytes expressing *osk::GFP C BRE*‾ alone (and thus strongly defective in translational activation), or in combination with a donor *osk* transgene as indicated. Late stage oocytes were used because the translational activation defect caused by the C BRE‾ mutations is not fully penetrant at earlier stages. Although only a small fraction of stage 9/10 *osk C BRE*‾oocytes have detectable Osk protein (**Reveal et al., 2010**), we wanted to remove this variability for experiments in which there could be small differences between different genotypes. Note that the imaging conditions used to provide these examples of different levels of rescue are not the same as those used for quantitation in panel **E**. The conditions for these images were chosen to highlight trace levels of GFP, and as a consequence the signal in some samples is saturated. (**D**) Levels of donor mRNAs. Values are in comparison to a rescuing *osk* transgene as in **Figure 2**. Error bars indicate standard deviations. (**E**) Posterior GFP fluorescence from *osk::GFP C BRE*‾ when expressed in combination with donor *osk* mRNAs bearing the indicated mutations. Each dot represents one late stage oocyte, with averages indicated by horizontal lines. All samples were from flies grown, fixed, processed and imaged in parallel (see 'Experimental Procedures' and **Figure 3—figure supplements 2** and **3** for details of quantitation). The student's t test was used to test for significance of differences relative to the *osk*+ donor mRNA (left), with the p values shown above.

The following figure supplements are available for figure 3:

**Figure supplement 1.** Predicted PTB-binding sites in the *osk* mRNA 3' UTR.

**Figure supplement 2.** Quantitation of fluorescence at the posterior pole of oocytes.

**Figure supplement 3.** Consistency of rescue of Osk::GFP expression.

**Figure supplement 4.** Qualitative differences in weak rescue of *osk::GFP C BRE*‾ expression.

**Figure supplement 5.** PT and dimer‾mutations in the *osk* mRNA 3' UTR do not disrupt posterior localization.

The *UAS-GFP* donor did not rescue the activation defect of *osk C BRE*‾ (**Figure 2D**, compare to **Figure 2A**). Similarly, the donor with the *osk* 3' UTR AB region provided no rescue. A donor carrying the *osk* 3' UTR C region, and thus the activation element mutated in the *osk C BRE*‾ recipient, provided a low level of rescue. Rescue was increased when both the AB and C regions were present, and the highest level of rescue was obtained when the *GFP* mRNA included the entire *osk* 3' UTR. The different levels of rescue cannot be attributed to differences in the amounts of the various donor mRNAs (**Figure 2E**).

The results of these experiments are consistent with 'regulation in *trans*'. The C region must be present to provide the missing translational activation element, while other Bru binding sites that do not contribute to activation (the AB region sites) are not sufficient. The weak rescue from the C region alone is consistent with the notion that rescue requires both the missing element and sequences which mediate assembly of *osk* RNPs. We suggest that by itself the C region supports weak association of *GFP-C* mRNAs with *osk C BRE*‾ transcripts, and this association is strengthened moderately by addition of the AB region and more strongly by addition of the full 3' UTR.

## *osk* RNP interactions are required for rescue in *trans*

To ask if RNP assembly is required for rescue in *trans*, we wished to inhibit or abolish each of the known forms of RNP assembly. Removing assembly factors - PTB and Bru - is not a good approach, as these proteins bind to many mRNAs and indirect effects on *osk* mRNA regulation are possible. In a more selective approach, the *osk* mRNA is modified, mutating elements involved in RNP assembly. With donor *osk* transgenes defective in one or more types of RNP assembly, any change in the ability to rescue the activation defect of the *osk::GFP C BRE*‾ recipient mRNA can be monitored.

Testing the role of Bru-mediated *osk* mRNA oligomerization (**Chekulaeva et al., 2006**) is problematic, as the Bru sites are also required to provide the missing regulatory element for *osk C BRE*‾

(above). However, the other contributions to assembly - mRNA dimerization and PTB-dependent RNP assembly - can be manipulated (*Figure 3A*). Direct *osk* mRNA dimerization requires a RNA element (*Jambor et al., 2011*), which is deleted in mutant *osk3'Δ665–685* (labeled as *dimer⁻* in *Figure 3*). Testing the role of PTB-mediated assembly could be done by mutating PTB-binding sites. These sites are spread throughout the *osk* mRNA 3' UTR (as shown by RNA binding assays), although individual sites have not been defined (*Besse et al., 2009*). However, because the binding specificity of *Drosophila* PTB is likely to be very similar or identical to that of the highly conserved mammalian homolog, for which binding sites have been characterized by a combination of approaches (*Pérez et al., 1997*; *Xue et al., 2009*), candidate sites can be identified (*Figure 3—figure supplement 1*). Mutating all the candidate sites (consisting of pyrimidine tracts or PTs) is the simplest strategy to disrupt PTB binding, but introducing so many mutations increases the probability of an inadvertent effect on rescue in *trans*. To strike a balance between these design considerations, a subset of the PT sequences was mutated (*osk3'PT^{mut}*)(*Figure 3—figure supplement 1*), focusing on those with features associated with the strongest sites for mammalian PTB binding, while avoiding mutations that might also disrupt *osk* mRNA dimerization or Bru binding. A UV crosslinking competition binding assay was used to determine if the PT mutations affected PTB binding (*Figure 3B*). PTB bound the radiolabeled *osk* mRNA 3' UTR probe mRNA (left lane). Increasing amounts of unlabeled *osk* 3' UTR RNA (*osk⁺*) effectively competed for binding. When the competitor RNA included the PT mutations (3'PTmut), competition was less effective, demonstrating that the mutations do affect PTB-binding sites.

We coexpressed different donor mRNAs (all present at similar levels; *Figure 3D*) together with the activation-defective *osk::GFP C BRE⁻* recipient mRNA, and measured posterior GFP levels in late stage oocytes to determine how effectively the activation defect was rescued (*Figure 3C*). The wild-type donor provides strong rescue of activation, as does the donor lacking the dimerization motif (*dimer⁻*)(*Figure 3E*). By contrast, the *osk3'PTmut* and *osk3'PTmut dimer⁻* donors provided almost no rescue of activation. Although no statistically significant difference between GFP levels with these latter two donors was detected, some oocytes had trace levels with the *osk3'PT^{mut}* donor, while none were above background with the *osk3'PT^{mut} dimer⁻* donor, consistent with an additive effect of the two types of mutations (*Figure 3E*). Examples of the trace posterior GFP signals with the *osk3'PT^{mut}* donor, and their absence with the *osk3'PT^{mut} dimer⁻* donor, are shown in *Figure 3—figure supplement 4*. The loss of rescue in *trans* cannot be attributed to failure of the donor mRNAs to localize to the posterior pole of the oocyte, as the PT mutations do not interfere with posterior localization (*Figure 3—figure supplement 5*).

Mutation of candidate PTB-binding sites in the *osk* mRNA clearly disrupts rescue in *trans*. Loss of PTB-dependent RNP assembly is one explanation of the results, and is consistent with 'regulation in *trans*'. Alternatively, inadvertent mutation of the binding site of some other factor could make the *osk3'PT^{mut}* mRNA a poor donor. To distinguish between these options two approaches were used.

For one approach, we reasoned that if *osk3'PT^{mut}* failed to rescue because of reduced PTB binding, this would likely be an additive defect that depends on mutation of multiple different PTs. Thus, mutating only subsets of the PTs might yield intermediate levels of rescue. By contrast, if loss of rescue was due to inadvertent mutation of a binding site for another factor, then, assuming that this other factor does not have the same binding specificity as PTB, this effect would likely be due to a specific PT mutation. If so, we would expect that only a single subset of PT mutations would disrupt rescue in *trans*.

The mutations in *osk3'PT^{mut}* were subdivided into four groups (*Figure 4A*, *Figure 3—figure supplement 1*). To confirm that each subset of PT mutations reduced PTB binding, short segments of the *osk* mRNA 3' UTR containing the subsets of mutations were tested in UV crosslinking assays. Each of the four 3' UTR segments bound PTB, and for each the binding was substantially lower when the RNA contained the PT mutations (*Figure 4B*).

The same subsets of mutations were also introduced into *osk* donor transgenes (*Figure 4A*, *Figure 3—figure supplement 1*). Just as for the *osk3'PTmut* mRNA, all the new *osk* donor transgenes were properly localized to the posterior pole of the oocyte (*Figure 3—figure supplement 5*). Each of these donors was impaired for rescue of the translational activation defect of the *osk::GFP C BRE⁻* recipient, although none was as defective as the *osk3'PT^{mut}* donor (*Figure 4D*). Thus, the PT mutations had an additive effect, and some subsets of the mutations inhibited rescue in *trans* more than others. The differences in rescuing activity did not correlate with the amounts of the different donor

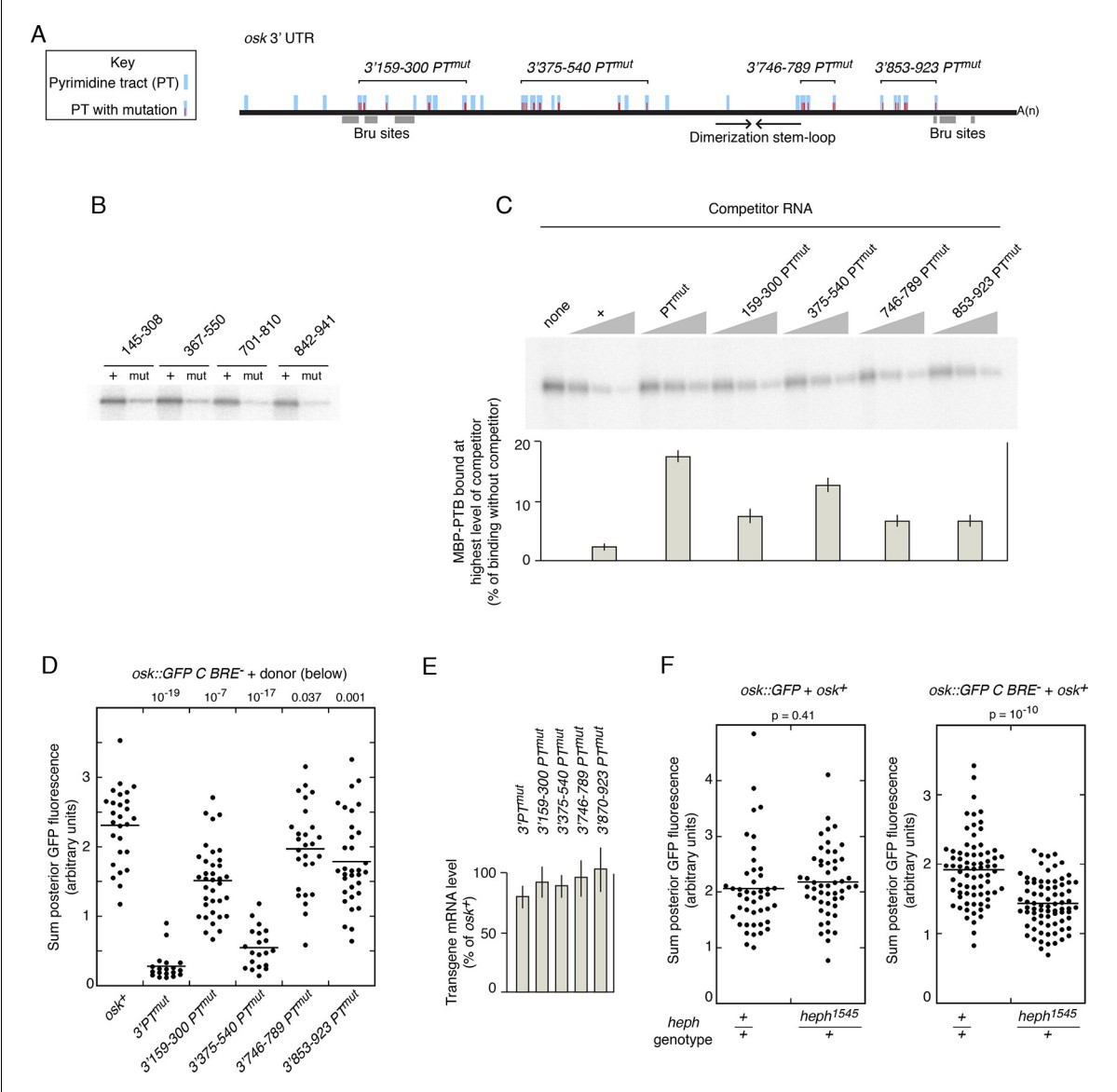

**Figure 4.** Evidence that PTB is required for rescue in *trans*. (**A**) Diagram of the *osk* mRNA 3' UTR. Above the black line (the 3' UTR) are indicated candidate PTB-binding sites (blue), consisting of any tract of mixed pyrimidines at least 4 nt in length. Mutations within pyrimidine tracts (PTs) that are present in the *osk* 3'PT*mut* transgene are indicated in red. Subsets of PT mutations, incorporated into genomic *osk* transgenes used in panel B, are indicated above (numbers correspond to positions in the *osk* 3' UTR). Below the black line, the Bru sites and position of the stem-loop structure containing the RNA dimerization motif are indicated. (**B**) Subsets of PT mutations disrupt PTB binding to short segments of the *osk* mRNA 3' UTR. Short segments of the *osk* 3' UTR, each encompassing a subset of PT sites noted in panel A, were used as RNA substrates for UV crosslinking assays with MBP-PTB. The extent of each RNA is indicated above (numbers correspond to positions in the *osk* 3' UTR). Radiolabeled RNAs were prepared in wild type form (+), or with the PT mutations from that region (i.e. the same mutations as indicated in red in A)(mut). (**C**) Subsets of PT mutations disrupt PTB binding to the complete *osk* mRNA 3' UTR. UV crosslinking assays with unlabeled competitor RNAs were performed as in *Figure 3E*, again using the complete *osk* 3' UTR as a radiolabeled-binding substrate. The competitor RNAs are indicated above, and consist of the complete *osk* 3' UTR with the designated PT mutations. The bar graph below shows the level of residual binding to the radiolabeled substrate RNA at the highest level of competitor (30x molar excess). Smaller values indicate stronger binding of the competitor RNA. Assays were performed three times, and the error bars indicate standard deviations. The 375–540 PT*mut* RNA was a significantly less effective competitor than the other RNAs with subsets of mutations (p<0.05 for all). (**D**). Posterior GFP fluorescence from *osk::GFP C BRE*⁻ when expressed in combination with donor *osk* mRNAs bearing the indicated mutations. Data are presented as in *Figure 3*. The student's t test was used to test for significance of differences relative to the *osk+* donor mRNA (left), with the p values shown above. Additional statistical tests evaluated effect size (Cohen's d) and power. Effect size and power, respectively, for the different donors relative to *osk+* were: *3'PTmut*, 5.01 and 1.0; *3'159–300 PT*ᵐᵘᵗ, 1.53 and >0.99; *3'375–540 PT*ᵐᵘᵗ, 4.12 and 1.0; *3'746–789 PT*ᵐᵘᵗ, 0.58 and 0.56; *3'853–923 PT*ᵐᵘᵗ, 0.88 and 0.91. The statistical significance of the reduced rescuing activity of the *3'746–789 PT*ᵐᵘᵗ donor is less compelling than for the other

*Figure 4 continued on next page*

Figure 4 continued

mutants, but this does not invalidate the overall conclusion that the PT mutations have an additive effect on loss of rescuing activity. (E) Levels of donor mRNAs. Values are in comparison to a rescuing *osk* transgene as in **Figure 2**. Error bars indicate standard deviations. (F) Posterior GFP fluorescence from *osk::GFP* (left) or *osk::GFP C BRE*⁻ (right) when expressed in combination with *osk*⁺. Flies were either *heph*⁺ or heterozygous for *heph*¹⁵⁴⁵, as indicated below. The student's t test was used to test for significance of differences between the two samples in each panel, with the p values shown above. Additional statistical tests evaluated effect size (Cohen's d) and power. Effect size and power, respectively, were 0.169 and 0.134 (left panel) and 1.12 and >0.99 (right panel).

mRNAs (**Figure 4E**). To determine if reduced donor activity displayed a correlation with reduced PTB binding activity, PTB binding was measured in competition-binding assays (**Figure 4C**). Binding of PTB to a radiolabeled wild type *osk* 3' UTR RNA was performed in the presence of increasing amounts of unlabeled *osk* 3' UTR competitor RNAs. Competitors with subsets of PT mutations had intermediate levels of competition, not as strong as the wild type *osk* 3' UTR but stronger than the *osk* 3' UTR with all PT mutations. Notably, among this group, the *osk* 3' UTR RNA with the 3'375–540 subset of PT mutations was the weakest competitor, just as the *osk3'375–540 PT*ᵐᵘᵗ transgene was the least effective donor for rescue in *trans*. Thus, we did observe a correlation between strength of PTB binding and strength of rescue in *trans*. For the other three subsets of PT mutations, their effects on PTB binding were too similar to one another to draw clear distinctions among them (**Figure 4C**), much like the effects of these mutations on rescue in *trans* (**Figure 4D**).

As a second approach to ask if PTB is required for rescue in *trans*, no PTB binding sites in either donor or recipient *osk* mRNAs were mutated, but instead the level of PTB protein was reduced. The *osk*⁺ donor and *osk::GFP C BRE*⁻ recipient mRNAs were coexpressed in flies either wild type or heterozygous mutant for *heph* (*heph* encodes PTB)(**Figure 4F**, right). Reducing the level of PTB activity in the *heph* heterozygotes impaired rescue of the translation activation defect, causing a lower level of Osk::GFP. By contrast, in a control experiment using a recipient mRNA with no translation defect, *osk::GFP*, reducing the level of PTB activity had no effect on Osk::GFP levels (**Figure 4F**, left).

Thus, the results of both kinds of tests to determine the role of PTB - mutating candidate PTB-binding sites or reducing PTB activity - point to the same conclusion: PTB-dependent assembly of *osk* mRNA into RNPs is required for rescue in *trans*, supporting the model of 'regulation in *trans*'.

## Mutation of candidate PTB-binding sites in *osk* mRNA affects the level of Gurken protein

If assembly of *osk* transcripts in RNPs promotes 'regulation in *trans*', would translation of other mRNAs present in the same RNPs also be influenced by the translational control elements in the *osk* mRNA? Candidates for such *trans* regulation are expected to share two features with *osk* mRNA: reliance on a shared factor for RNP assembly, and overlap in subcellular distribution. Because PTB is most strongly implicated in rescue of *osk* mRNA regulatory defects in *trans*, we focused on mRNAs with candidate PTB-binding sites.

Among ovarian mRNAs, one of the most highly enriched for candidate PTB-binding sites is *gurken* (*grk*) (**Supplementary file 1**). Furthermore, PTB associates with *grk* mRNA, both in vitro and in vivo (**McDermott et al., 2012**; **McDermott and Davis, 2013**). Notably, *grk* mRNA is, like *osk*, highly enriched in the oocyte during previtellogenic stages (**Figure 5A,B**) (**Neuman-Silberberg and Schüpbach, 1993**). Later, at stage 8, both mRNAs align at the anterior margin of the oocyte, but then go their separate ways within the oocyte: *grk* becomes restricted to the anterodorsal corner, while *osk* moves to the posterior (**Kim-Ha et al., 1991**; **Neuman-Silberberg and Schüpbach, 1993**; **Ephrussi et al., 1991**). Thus, *grk* and *osk* mRNAs have similar distributions for part of oogenesis and have the potential to reside in the same RNPs during that period. Consequently, each mRNA could influence the translation of the other. Because translation of *osk* mRNA is strongly repressed for the entire period of shared localization, the most likely *trans* effect is for *osk* mRNA to confer some degree of translational repression on *grk* mRNA.

To address the possibility that *osk* transcripts influence *grk* mRNA translation, we monitored Grk protein levels in the oocytes of previtellogenic stage 5/6 egg chambers (**Figure 5C**). If wild-type *osk* mRNAs are present (**Figure 5D** upper), then 'regulation in *trans*' is possible. If *osk3' PT*ᵐᵘᵗ mRNA is the only *osk* mRNA present (**Figure 5D** lower; loss of PTs is indicated by fewer yellow dashes), then

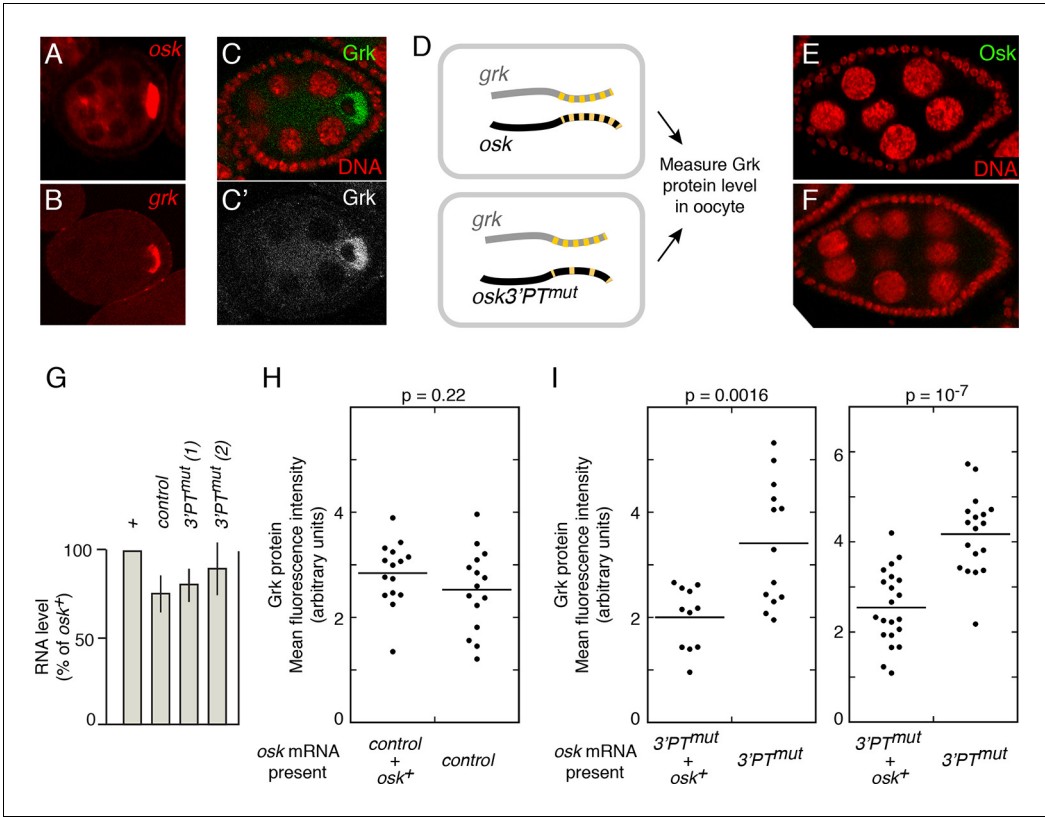

**Figure 5.** *osk* mRNA influences the level of Grk protein. **A** and **B**. Distribution of *osk* (**A**) and *grk* (**B**) mRNAs in egg chambers early in oogenesis. mRNAs were detected by in situ hybridization. For each egg chamber, the cell at right with intense signal is the oocyte. (**C**) Distribution of Grk protein at stage 6 of oogenesis. **C** shows both Grk (green) and nuclei (red), and **C'** shows only Grk (white). Grk is most strongly expressed in the oocyte, and is excluded from the oocyte nucleus. (**D**) Experimental design. Each rectangle represents an oocyte, showing the *grk* and *osk* mRNAs present. Both have multiple PTs, indicated by the yellow dashes. The *osk 3'PT^{mut}* mRNA has many PTs mutated, and thus fewer yellow dashes. (**E** and **F**) Immunodetection of Osk protein, showing no detectable precocious expression of Osk protein from either the control *osk* mRNA (**E**) or the *osk3'PTmut* mRNA (**F**). Nuclei are red. (**G**) Levels of donor mRNAs. Values are in comparison to a rescuing *osk* transgene as in *Figure 2*. Error bars indicate standard deviations. (**H**) Levels of Grk protein in oocytes when expressed in the presence of both wild type and control *osk* mRNAs (left) or only the control *osk* mRNA (right). The control *osk* mRNA is from the *osk 11-13^-* transgene (*Figure 6*) which has no defect in Osk expression or rescue in *trans*. See 'Experimental Procedures' and *Figure 5—figure supplement 1* for details of quantitation. (**I**) Levels of Grk protein in oocytes when expressed in the presence of *osk* mRNAs as indicated at bottom. In the left panel, *osk 3'PT^{mut}* line 1 was used, and in the right panel *osk 3'PT^{mut}* line 2 was used. Additional statistical tests evaluated effect size (Cohen's d) and power, respectively: 1.51 and 0.94 (left panel) and 1.91 and >0.99 (right panel). For comparison, the values for the control experiment in panel H were 0.45 (effect size) and 0.22 (power).

The following figure supplement is available for figure 5:

**Figure supplement 1.** Quantitation of fluorescence in Stage 5/6 oocytes.

---

any PTB-dependent association of *osk* and *grk* mRNAs should be impaired, and 'regulation in *trans*' could be reduced. Remarkably, the type of *osk* mRNA present did affect Grk protein levels. When only *osk3'PT^{mut}* mRNA was present, Grk protein levels were higher than when *osk^+* mRNA was also present (*Figure 5I*). This effect was observed in each of two separate experiments, making use of different lines of the *osk3'PT^{mut}* transgene. As a control, we performed the same type of experiment with an *osk* transgene that is a strong donor for rescue of *osk* mRNA regulatory defects and found no change in Grk protein levels (*Figure 5H*). The differing behaviors of the *osk3'PT^{mut}* and *osk* control transgenes cannot be attributed to differences in mRNA levels (*Figure 5G*). Importantly, none of

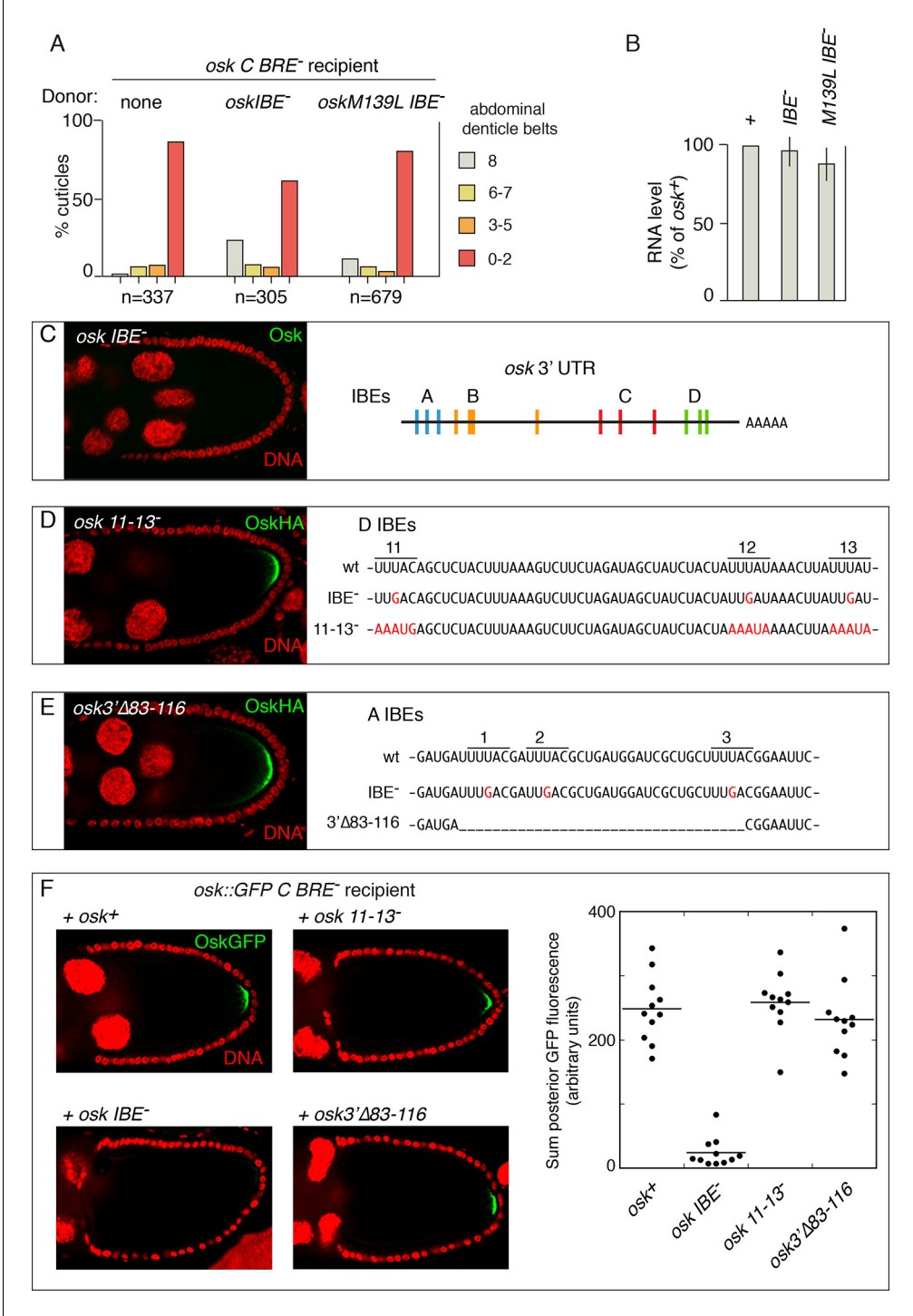

**Figure 6.** IBE mutations disrupt rescue in *trans* and have gain-of-function properties. (**A**) Embryonic patterning assays to monitor rescue of the translational activation defect of *osk C BRE⁻*. (**B**) Levels of transgenic *osk* mRNAs. Values are relative to an *osk+* transgene that fully rescues an *osk* mutant. Error bars indicate standard deviations. (**C**) Absence of Osk expression from the *osk IBE⁻* transgene (left)(stage 9 egg chamber) and positions of the IBEs in the *osk* 3' UTR [diagram adapted from Figure 6A of (***Munro et al., 2006***)]. The *osk IBE⁻* transgene used has the A subset of mutations, and makes no detectable Osk protein (the same phenotype as *osk* mutants with the C or D subsets of IBE⁻ mutations) (***Munro et al., 2006***). (**D**) Mutation and phenotype of IBEs 11–13. Left, Osk expression from the *osk 11-13⁻* transgene mRNA (stage 9 egg chamber). The *osk 11-13⁻* transgene has an HA epitope tag, which was used for immunodetection (an *osk⁺* transgene with the same tag is expressed at similar levels; not

*Figure 6 continued on next page*

*Figure 6 continued*

shown). Right, sequences of the *osk* 3' UTR region containing IBEs 11–13. Mutated bases are in red. The IBE⁻ mutations are those used by *Munro et al. (2006)*. (E) Mutation and phenotype of IBEs 1–3. Left, Osk expression from the *osk3'Δ83–116* transgene mRNA (numbering indicates position in the *osk* mRNA 3' UTR)(stage 9 egg chamber). Right, sequences of the *osk* 3' UTR region containing IBEs 1–3. Mutated bases are in red, and the region deleted is indicated by underscoring. The IBE⁻ mutations are those used by *Munro et al. (2006)*. (F) Osk:: GFP expression in stage 10 egg chambers (posterior portion only) from females expressing *osk::GFP C BRE⁻* in combination with the donor *osk* mRNAs indicated. Fluorescence intensities were quantitated for the graph at right.

the *osk* mRNAs in these experiments produce any Osk protein at these stages of oogenesis (*Figure 5E,F*). Therefore, we conclude that *osk* mRNA, not Osk protein, influences Grk expression. The increase in Grk levels when *osk* mRNA is impaired in PTB-dependent RNP assembly is consistent with the strong translational repression of *osk* mRNAs acting in *trans* to confer some degree of translational repression on *grk* mRNAs present in the same RNPs.

## The idiosyncratic *osk IBE⁻* mRNA

As described above, we found that each of several *osk* mutants with defects in 3' UTR regulatory elements could be rescued in *trans*, consistent with the notion that rescue relies on long-range interactions that bridge between donor and recipient mRNAs. However, the *osk IBE⁻* mutant is an exception to this rule. The IBE sequence - UUUAY (Y is pyrimidine) - was identified as a binding site for Imp protein (*Munro et al., 2006*). Thirteen copies of the IBE appear in the *osk* mRNA, all in the 3' UTR. The *osk* IBEs were mutated in subsets to test their role. Three of the mutants (subsets A, C and D; *Figure 6C*) fail to accumulate Osk despite having normal localization of *osk* mRNA, while the fourth mutant (subset B) has normal Osk expression. Mutating the A, C, or D subsets of IBEs disrupts colocalization of Imp protein with *osk* mRNA at the posterior pole of stage 9/10 oocytes, supporting the notion that Imp binds *osk* mRNA through these elements and mediates activation of *osk* mRNA translation (*Munro et al., 2006*). However, mutants lacking Imp protein have normal *osk* expression, suggesting that the IBEs also bind another factor, with this second factor contributing to activation of *osk* translation (*Munro et al., 2006*; *Geng and Macdonald, 2006*).

Previously, we found that the *osk IBE⁻* mutant was not strongly rescued in *trans* (*Reveal et al., 2010*). Now, we asked if the *osk IBE⁻* mutant can act as a donor for rescue in *trans*. Neither of two transgenes with mutated IBEs strongly rescued the translational activation defect of the *osk C BRE⁻* mutant (*Figure 6A*). Similarly, neither IBE mutant restored protein expression of the *osk::GFP C BRE⁻* mutant (*Figure 6F*, data not shown). Failure to rescue strongly was not due to insufficient mRNA, as both IBE mutant *osk* mRNAs were present at high levels (*Figure 6B*). Thus, mutation of the IBEs disrupted rescue in *trans*, with the mutant *osk* mRNAs unable to act as either donor or recipient.

Several features of the IBE mutants, notably their inability to participate in rescue in *trans*, as well as a report that Imp protein has only weak affinity for *osk* mRNA (*Geng and Macdonald, 2006*), prompted further analysis of the IBEs. We made two new mutants (*Figure 6D,E*), each affecting a subset of IBEs implicated in activation of *osk* translation (*Munro et al., 2006*). In mutant *osk 11-13-*, the 3'-most IBEs (subset D) were altered. Instead of mutating the central position in the UUUAY pentamer from U to G (as in the original *osk IBE⁻* mutants), each nucleotide of each pentamer was changed, and so a stronger defect in binding - to Imp or to the unknown factor - is expected. Notably, the *osk 11-13-* mutant had normal Osk accumulation (*Figure 6D*). For mutant *osk 3'Δ83–116*, the entire region encompassing the 5'-most IBEs (subset A) was deleted. Again, this mutant is expected to more strongly disrupt Imp/unknown binding than the corresponding *osk IBE-* mutant, but it had normal Osk accumulation (*Figure 6E*). Thus, the original IBE mutations appear to disrupt translational activation of *osk* mRNA not by removing a binding site, but by creating a novel type of element, such as a binding site for another factor.

The new mutants lacking subsets of IBEs were also tested for the ability to act as donors for rescue in *trans*. The activation-defective *osk:GFP C BRE-* was used as the recipient, and we monitored GFP to determine if activation was rescued. The original *osk IBE-* mutant did not substantially rescue translation of the recipient mRNA. By contrast, both *osk 3'Δ83–116* and *osk 11-13-* mutants strongly

rescued translational activation (*Figure 6F*). It seems likely, then, that the gain-of-function property of the original IBE⁻ mutations not only disrupts *osk* mRNA translation, but also interferes with the ability of the mRNA to participate in rescue in *trans*.

## Discussion

Long-range interactions involving structures or factors associated with the different ends of an individual mRNA are required for translation. Such interactions are not inherently intra-transcript, and a factor bound to the 5′ end of one mRNA could engage a factor bound to a 3′ region of another mRNA. Nevertheless, inter-transcript interactions have generally not been considered in models of translation or its regulation. The *osk* mRNA is an exception, displaying the notable property of rescue of regulatory mutations in *trans*. We suggested that this phenomenon results from inter-transcript interactions, with the participating mRNAs placed in close proximity in RNPs to facilitate the interactions. The work described here establishes parameters for this rescue in *trans*, shows that conditions which reduce RNP assembly diminish rescue in *trans* and provides evidence that regulation imposed on the *osk* mRNA influences the expression of *grk* mRNA by a similar *trans* effect. Our results are fully consistent with and lend considerable support to the model of 'regulation in *trans*'.

We have now asked which regulatory defects in the recipient mRNAs can be rescued in *trans*, and what features are required in the donor mRNAs to allow rescue. Learning more about the underlying rules can provide support for, or argue against, the model of 'regulation in *trans*'.

We first consider the recipient mRNAs. In the context of the proposed model, mutations affecting long-range regulatory interactions should be susceptible to rescue in *trans*, while other types of regulatory defects would not. Testing new regulatory mutants as recipients, we found that three mutants with mutations in the 3′ UTR could be rescued in *trans*, while a mutant with defects in the 5′ region of the mRNA could not. Sensitivity of the 3′ UTR regulatory elements to rescue in *trans* is consistent with their participation in long-range interactions to activate translation. In particular, the *osk3′1004–1008* mutant is defective in an A-rich element, which likely serves to bind PABP (*Vazquez-Pianzola et al., 2011*). PABP interacts with the translation initiation factor eIF4G (*Tarun et al., 1997*), associated with the mRNA 5′ cap, and this interaction has the potential to occur between different transcripts. As for the 5′ region mutant which is not rescued in *trans*, why activation of translation is defective is not understood (*Kanke and Macdonald, 2015*), but the position of the mutations suggests a local, rather than long-range, effect on initiation of translation.

Experiments addressing the requirements for an mRNA to act as a donor in rescue in *trans* lead to several conclusions. First, we find that the element missing from the recipient must be provided by the donor, as predicted by the model of 'regulation in *trans*' [but also consistent with other models - see (*Wilusz and Wilusz, 2010*)]. Second, although the missing element is required in the donor, it is not sufficient, and additional 3′ UTR sequences must be present to obtain a substantial degree of rescue. These results are also consistent with 'regulation in *trans*', with the additional sequences presumably acting to promote interactions between the different transcripts and thus allow the missing regulatory element to act in *trans*. Third, many *osk* mRNAs can act as donors for rescue in *trans*; this appears to be the default property. Even if an *osk* mRNA is not receptive to rescue in *trans*, this does not exclude the possibility that it can act as a donor. For example, the *oskM1RΔ121–150* mutant cannot be rescued in *trans* but is a strong donor. Again, these observations are consistent with 'regulation in *trans*', in which the donor mRNA need not itself be translated, it needs only to contain the regulatory element missing from the recipient as well as sequences that mediate association of the different mRNAs. Fourth, there are mutations which interfere with the ability of an *osk* mRNA to act as a donor for rescue in *trans*. These fall into two groups, the enigmatic IBE mutant, and the mutants designed to disrupt RNP assembly.

The IBE mutant is completely defective in activation of *osk* mRNA translation (*Munro et al., 2006*) and cannot be rescued in *trans*. As a donor, the IBE mutant confers only very weak rescue of other regulatory defects. New mutants targeting the IBEs, which more substantially alter or eliminate the IBE motifs, have no regulatory defects and are strong donors for rescue in *trans*. Thus, the original IBE mutations appear to cause a dominant gain-of-function defect, perhaps by serendipitous formation of a binding site for an unknown factor. Why do the original IBE mutations in one subset of IBE motifs, the B group, have no consequences? Notably, the B group IBEs are embedded in a cluster of Bru-binding sites. We suggest that bound Bru could occlude binding of the unknown factor,

thus avoiding the adverse effects. In any event, the original IBE mutants are clearly unusual and need to be considered separately from mutants which disrupt bona fide regulatory elements. Learning how the IBE mutant inhibits rescue in *trans* may well provide important insights into the mechanism, and will be the focus of future work.

Finally, mutants designed to disrupt RNP assembly also fail to rescue in *trans*. Mutation of many PTs in the donor mRNA most dramatically disrupted rescue in *trans*, with lesser defects from mutating subsets of these PTs. The additive nature of the PT mutations is fully consistent with the notion that they serve as binding sites for PTB, which assembles *osk* mRNA into RNPs in vitro and is required for association of *osk* mRNAs in vivo (*Besse et al., 2009*). As an independent test of the role of PTB in rescue in *trans*, we also monitored the effects of reducing PTB gene dosage when the donor and recipient *osk* mRNAs had intact PTB-binding sites. Consistent with 'regulation in *trans*', lowering the activity of PTB diminished rescue of the translational activation defect, leading to a lower level of Osk protein. PTB is implicated in translational repression of *osk* mRNA (*Besse et al., 2009*), and so a reduced level of PTB could also influence Osk protein levels more directly. However, a lower efficiency of repression would increase, not lower, the level of Osk protein, and the effect we observe is more reasonably attributed to 'regulation in *trans*'.

By the model of 'regulation in *trans*', almost any mRNA should be susceptible to inter-transcript interactions influencing its translation. In practice, such effects would be most likely for mRNAs in close proximity to one another, either because of an overall high mRNA concentration or because of coassembly in RNPs. To consider this possibility, we focused on the *grk* mRNA, which shares with *osk* mRNA patterns of subcellular distribution as well as association with PTB. Remarkably, removing candidate PTB-binding sites from *osk* mRNA led to elevation of Grk protein levels early in oogenesis. Because translation of *osk* mRNA is strongly repressed at this developmental stage, association of the *osk* and *grk* mRNAs could lead to some degree of translational repression of *grk*. Disrupting this association would reduce repression of *grk*, leading to higher Grk protein levels, as we observed. It is also possible that reducing the amount of PTB bound to *osk* mRNA would increase the pool of PTB available for binding to *grk*. However, if this occurs, given the evidence that PTB acts in the cytoplasm as a repressor of translation (*Besse et al., 2009*), the expected outcome would be enhanced repression of *grk* mRNA and less Grk protein, not more.

Our evidence that translational regulation can occur in *trans*, with inter-transcript interactions affecting the translation of participating mRNAs, raises the question of whether this phenomenon is unusual, or more widespread but unappreciated. Unless a mechanism exists to specifically exclude inter-transcription interactions, there is no compelling reason why such interactions would not occur, and their probability may be dictated by mRNA concentrations. The absence of other reports of this phenomenon may be misleading, as detecting *trans* effects is not straightforward. For *osk* mRNA, rescue in *trans* was discovered by comparing the effects of *cis* regulatory mutations in the presence or absence of other *osk* transcripts (*Reveal et al., 2010*). To the best of our knowledge, this approach is not common. Furthermore, it requires an RNA null allele for the regulated gene, a reagent not always available. Despite the absence of direct tests for rescue in *trans*, recent advances in understanding how mRNAs are localized may be relevant. Often, localized mRNAs are subject to translational regulation to ensure that the protein is only synthesized at the appropriate time or location (*Jung et al., 2014*). Curiously, several such mRNAs - but not *osk* - are transported in RNPs containing a single mRNA molecule (*Batish et al., 2012*; *Amrute-Nayak and Bullock, 2012*; *Mikl et al., 2011*; *Little et al., 2015*). Perhaps, such isolation of individual mRNAs serves to ensure that 'regulation in *trans*' does not occur and that regulation is constrained to that imposed by embedded *cis* elements.

## Experimental procedures

### Flies and transgenes

Mutants at the *osk* locus: *osk^{54}* (*Lehmann and Nüsslein-Volhard, 1986*), *osk^{A87}* (*Jenny et al., 2006*), *Df(3R)osk* (*Reveal et al., 2010*), and *osk^{0}* (*Kanke et al., 2015*). For experiments in which an *osk* transgene was tested as the only source of *osk* mRNA, the background was an *osk* RNA null combination, either *osk^{A87}/Df(3R)osk*, *osk^{A87}/osk^{0}*, *osk^{0}/Df(3R)osk*, or *osk^{0}/osk^{0}*. We have not observed any differences in phenotypes from use of the different *osk* RNA null combinations. Genomic *osk* transgenes were based on a genomic fragment that fully rescues *osk* null mutants (*Kim-Ha et al.,*

*1991*). The genomic *osk::GFP* transgenes include the complete *osk* sequence, with *mGFP6* (*Haseloff, 1999*) introduced at position T140 (one codon after the initiation codon for Short Osk). Except for *osk IBE⁻* (*Munro et al., 2006*) and *osk C BRE⁻* (*Reveal et al., 2010*), all genomic *osk* transgenes lacking GFP have three copies of the HA epitope tag at position T140; this tag has no detectable effect on *osk* activity (*Kanke and Macdonald, 2015*; *Jones and Macdonald, 2015*; *Kim et al., 2015*). IBE⁻ transgenes have the A subset (*Munro et al., 2006*) of IBE mutations. Mutations were introduced by PCR, or from synthetic DNA fragments (gBlocks; Integrated DNA Technologies, Coralville, Iowa). *UAS-GFP* transgenes, with or without parts of the *osk* 3′ UTR, are those used previously (*Reveal et al., 2010*), or are identical but differ in the portions of the *osk* 3′ UTR included. The *matalpha4-GAL-VP16* driver (*Martin and St Johnston, 2003*) was used for expression of *UAS-GFP* transgenes.

## Analysis of embryonic axial patterning

Female flies of the genotype to be tested, together with sibling males, were aged in well-yeasted vials for 3–4 days after eclosion, and then transferred to small population cages for egg laying on yeasted apple juice plates. The plates were changed at intervals (∼12–24 hr), the embryos aged for a further 20–24 hr, and cuticles were prepared for mounting in Hoyer's medium (*Wieschaus et al., 1986*). A wild-type embryo has eight abdominal ventral denticle belts. When *osk* activity is reduced, fewer denticle belts form with the number of belts proportional to the level of *osk* activity. Conversely, if there is excess *osk* activity, anterior structures are lost. For the transgenes studied here, only reduced *osk* activity was detected. Each embryo was scored for the number of complete abdominal denticle belts. The results presented in *Figs. 1B*, *2A* and *6A* represent the embryos from three or more sequential embryo collections. For the results of *Figure 2D*, the results from two separate experiments (i.e. independent sets of crosses to generate the flies) were combined to increase sample sizes.

## Detection of RNAs and proteins

RNA levels were determined by RNase protection assays, using *rp49* for normalization of sample amounts (*Reveal et al., 2010*). At least three assays were done for each mRNA. RNA distributions were detected by in situ hybridization (*Jambor et al., 2011*).

Proteins were imaged in whole mount samples by confocal microscopy with a Leica TCS-SP (*Reveal et al., 2010*; *Kim-Ha et al., 1995*). Antibodies for immunodetection: rabbit anti-Osk (*Reveal et al., 2010*), 1/3000; mouse anti-HA (Covance), 1/300; mouse anti-Grk 1D12 [Developmental Studies Hybridoma Bank (DSHB)], 1/10; and mouse anti-Hts 1B1 (DSHB), 1/1. TO-PRO-3 Iodide (Invitrogen)(1:1,000) was used to stain nuclei. For immunodetection, and for detection of GFP from fusion proteins, samples in comparison groups (corresponding to sets of data in outlined boxes in *Figures 3–5*) were from flies grown in parallel and aged in well-yeasted vials. They were dissected, fixed, and stained in parallel, and imaged in a single session using identical confocal settings in which the fluorescence signals were not saturated.

For quantitation of posterior Osk::GFP in late stage oocytes, we first determined how reliably an optical section could be visually identified as having the maximum signal intensity. Note that all late stage oocytes, having a high anterior/posterior to dorsal/ventral axial ratio, are positioned with the anterior/posterior axis parallel to the plane of the slide. This orientation ensures that a z series of images is comparable for all oocytes. A z position appearing to have the highest level of Osk::GFP signal was noted, and then a z series of images roughly centered on that position was collected. Signal intensities were determined (as described below), and plotted across the z dimension. As shown in the two oocyte examples in *Figure 3—figure supplement 2*, this approach was adequate to identify a z position having a signal intensity within 2% of the maximum measured intensity. To measure signal intensity of the posterior crescent of Osk::GFP, the posterior peripheral region of each oocyte was traced in FIJI using a Wacom Intuos tablet. Because oocytes differ in shape, a single standardized trace outline could not be used. When the Osk::GFP signal intensity was strong, the region to be traced was readily identified, with the boundaries of the traces extending well beyond the anterior-most signal. When Osk::GFP signal intensity was low or undetectable, a region similar in shape and size was traced. Within the traced region for each oocyte, the total signal was measured. Other options for quantitation of the posterior crescent of Osk::GFP are to determine average or maximum

signal intensity. Neither provides a reliable estimate of total Osk::GFP. For an average intensity to be reliable, the area scored must be identical for all oocytes. Given the variation in oocyte shape, tracing areas identical in size is not practical. For a maximum signal intensity to be reliable, there must be a point source of Osk::GFP. This is not the case. To evaluate the consistency of the assay, a single biological sample was divided into two portion, and each was mounted on a separate slide. Both slides were imaged in a single session. Quantitation using the approach described showed a high level of consistency (*Figure 3—figure supplement 3*).

Quantitation of Grk protein in early oocytes presents different challenges and a different approach was used. Within early oocytes Grk protein is roughly uniform in distribution, except that the protein is excluded from the nucleus. Because all samples in these experiments contain readily detectable Grk, the position of the nucleus (unstained) is obvious and the cytoplasm can be easily traced. Given the uniform distribution of Grk in the cytoplasm, measuring an average signal intensity is appropriate. By contrast, measuring the total amount of signal does not provide a reliable estimate of Grk levels, as the area of the oocyte varies substantially depending on z position. For these images, additional steps were taken to obtain similar data from all samples. First, only egg chambers oriented in a roughly horizontal position were analyzed. This ensures that the position of the nucleus (lacking Grk), and the region of the oocyte it occupies, are similar for all samples. Second, the focal plane was chosen to include the oocyte nucleus, thus avoiding the peripheral regions of the oocyte. Near the periphery, only a small portion of the oocyte is included in an optical section, which causes high variability in average signal intensity. Limiting the analysis to the more central region yielded only minor variation in measured signal intensities (*Figure 5—figure supplement 1*), much less than the differences observed in the experiments of *Figure 5*.

## RNA binding

A maltose-binding protein-PTB fusion protein (*Besse et al., 2009*) was expressed in *E. coli* BL21/ pLysS and affinity purified on amylose resin (New England Biolabs) according to the manufacturers protocol. UV crosslinking assays were performed as described (*Macdonald et al., 1995*). Radiolabeled RNAs were synthesized by in vitro transcription (MaxiScript; Ambion) with alpha 32-P UTP from plasmid templates containing either the full *osk* mRNA 3' UTR (*Figure 3* and *Figure 4C*), or from plasmid templates containing short segments of the *osk* 3' UTR (*Figure 4B*). Competition-binding experiments were performed as described (*Macdonald et al., 1995*). For all assays, phosphorimaging of dried gels was used to visualize and quantitate binding. All assays were performed at least three times, with no significant variation in results.

## Statistical analysis

The patterning activity and imaging assays can be divided into two groups: those with dramatically different outcomes and those with graded differences which can be small. For the first group, such as the patterning assays in *Figure 2A* and the imaging results in *Figure 2C*, no statistical analysis is included. For the second group, statistical tests were used to evaluate the significance of any observed differences. For comparisons between two samples, the student's t test was used, with p values provided in each figure. Determinations of effect size and power (*post hoc* calculation) were performed using G*Power (*Faul et al., 2007*; *Faul et al., 2009*). To facilitate comparisons in patterning assays of *Figure 2D*, embryos in different phenotypic classes were assigned numbers of abdominal denticle bands as follows: 6–7, 6.5; 3–5, 4; and 0–2, 1 (embryos scored with 8 bands retained the 8 value). The assigned values were used to calculate average number of denticles bands and standard deviations, from which effect size and power were determined in G*Power. These values were also used in student's t tests.

## Acknowledgements

We thank members of the Fischer, Macdonald and Stein labs for comments during the course of the work, Sally Amen of the University of Texas at Austin Department of Statistics and Data Sciences for advice on statistical analysis, Anne Ephrussi for antibodies and the PTB expression plasmid, and Janice Fischer for comments on the manuscript. The 1D12 and 1B1 monoclonal antibodies were obtained from the Developmental Studies Hybridoma Bank developed under the auspices of the NICHD and maintained by The University of Iowa, Department of Biology, Iowa City, Iowa 5224.

# Additional information

## Funding

| Funder | Grant reference number | Author |
|---|---|---|
| National Institute of General Medical Sciences | R01 GM096730 | Paul M Macdonald |

The funders had no role in study design, data collection and interpretation, or the decision to submit the work for publication.

## Author contributions

PMM, Conception and design, Acquisition of data, Analysis and interpretation of data, Drafting or revising the article; MK, AK, Acquisition of data, Analysis and interpretation of data, Drafting or revising the article

## Author ORCIDs

Paul M Macdonald, http://orcid.org/0000-0001-5993-5343

# Additional files

## Supplementary files

• Supplementary file 1. Lists of *Drosophila* genes expressed in oogenesis or in oocytes were obtained from Flybase.org using QuickSearch and searching for Stage: oogenesis or Tissue: oocyte. Either the entire mRNA, or just the 3′ UTR, of the longest variant of each mRNA was scanned for the pyrimidine tracts indicated.

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
