## [Decision Letter]

Thank you for submitting your work entitled "Community effects in regulation of translation" for consideration by *eLife*. Your article has been reviewed by two peer reviewers, and the evaluation has been overseen by a Reviewing Editor and K VijayRaghavan as the Senior Editor.

The reviewers have discussed the reviews with one another and the Reviewing editor has drafted this decision to help you prepare a revised submission.

Summary:

Both reviews on the manuscript have been completed and there is general concordance on several points. First, both reviewers feel that the work is of broad interest in taking earlier observations by your group showing that BREs' can be rescued in *trans* by another mRNA transcript, and extending these observations to show that other *osk* mutations can also be rescued in a manner potentially dependent on the PTB protein (and its binding sites). And, while both reviewers are inclined to think that indeed this idea is correct (if not the complete story), both reviewers feel that additional experiments are needed to support this specific claim. Most importantly, the distribution of the mutant RNA needs better characterization in the presence and absence of PTB (i.e. in the presence of PTB binding site mutations and with diminished PTB levels). One of the reviewers proposes the use of FISH to help support the proposed mechanism – two color FISH should help correlate the observed regulatory effects with colocalization. The second reviewer proposes some IP style experiments to address a similar point (where the amount of RNAs recovered correlate with the predicted strengths of the PTB binding sites, etc.). There were other experiments proposed to ask whether the nature of the mRNA interaction was reciprocal, and whether co-regulation happens in both directions. The reviewers agree that this would be interesting, but perhaps beyond the scope of the present work.

In light of the broad agreement of the reviewers, we would be happy to receive a revised manuscript should there be additional experiments provided to support the proposed model that PTB binding is critical for both colocalization and coregulation, demonstrated more directly through biochemistry/visualization, and not simply through genetic approaches.

*Reviewer #1:*

This manuscript is a follow up of the previous 2010 Developmental Cell paper (Dual Functions of BREs in Translational Regulation), the authors found that mutation of *osk* 3' UTR Bruno Recognition Element (BRE) results in developmental defects (measured in the number of cuticles in embryos). Coexpression of a second, protein null *osk* transcript (*osk^54^*) is able to rescue the developmental defects.

The authors carried out more mutations in the donor and the recipient *osk* mRNAs to try to understand what factors/elements are important for the rescue in *trans*. The authors characterized developmental defects and used GFP reporters to monitor rescue. There are two new conclusions from the paper. (1) PTB dependent mRNP assembly is essential for rescue in *trans*. (2) The translation regulation can affect other RNAs in the same RNP. Both of these are consistent with their in *trans* regulation model. The authors also perform an experiment on the IMP binding element mutant, which is not very conclusive. The second conclusion is novel and shows a potential physiological role of *trans* regulation. However, there are a few elements worth further clarification, especially with the first conclusion.

1) By deletion of nt121-150, the mutant cannot be rescued (Figure 1). The authors conclude that the 5' UTR is not important for being-rescued in trans. Since the deletion not only disrupts the long form Osk protein, but also the 5' UTR of short form *osk* RNA, the transcript (D121-150) itself might have problems for translation regulation. In order to make a strong statement that the 5' UTR is not important for rescue of a recipient mRNA, additional mutations or experiments will be needed.

2) It looks like that the RNP assembly is crucial for *trans* rescue. The mechanism proposed is that the donor and the acceptor mRNA need to be packaged together into an mRNP for this to happen. The authors showed development defects with GFP reporter expression. However, many mutations might result in defects in RNA localization or RNA assembly. Therefore, the distribution of the mutant RNA needs better characterization, for example, by FISH to donor and recipient, not just by qPCR to measure the total amount.

3) The authors argue that PTB dependent RNP assembly is crucial for the *trans* rescue. It is likely that the RNP assembly is necessary for the rescue. There are many ways that RNP assembly could be disrupted by the mutants. The point is how to prove it is PTB dependent. The authors mutated a set of PTB binding sites and showed rescue defects. There are two points the authors might want to show. First, these mutations may result in no or decreased PTB binding. The second might be hard to prove: so many mutations might disrupt something else. The authors also used a heterozygous mutant PTB to reduce the PTB level (is that true?). But as the author notes, PTB is essential for many RNA functions and might have indirect effects on *osk*.

4) The inclusion of data from IBE mutations is puzzling and doesn't help the manuscript. It is idiosyncratic, as the authors suggest. I would remove it.

Recommendation:

The novelty of the paper lies in showing a mechanism for rescue in *trans*. The authors provide data that PTB is part of this mechanism but this requires further work. Part of the problem is that they rely almost exclusively on genetics, which leads to uncertainty about epistatic mechanisms leading to their observations. For instance, the increase in Gurken is interesting, but phenomenological. While the proposed mechanism of rescue is likely correct, it needs validation before the manuscript is acceptable. FISH experiments may help to support the proposed mechanism. Two color FISH will help correlate rescue with colocalization. Furthermore, this will show the importance of homotypic clustering in the granules (an approach recently described by the Lehmann lab, 2015, Nature Communications). If the FISH experiments yield the predicted result, they will suggest a mechanism where homotypic interactions are necessary for rescue. As the manuscript now stands, it is not sufficiently novel or definitive for *eLife*.

*Reviewer #2:*

In this manuscript, Macdonald et al. have provided strong evidence in support of a model of post-transcriptional regulation in *trans*. Through their study of *oskar* and *gurken* mRNAs, the authors have characterized the regulatory elements required on donor and recipient mRNAs to allow for such regulation to take place. The authors go on to show that mutations in candidate PTB binding sites in the donor mRNA abolish the ability of the donor to regulate the recipient mRNA, and that regulation in *trans* is dependent on PTB dosage. The authors generalize their model by showing that *oskar* mRNA is capable of regulating another mRNA with which it co-localizes, *gurken*, in a manner that once again requires candidate PTB binding sites on the donor. They conclude that PTB-dependent particle assembly mediates regulation in *trans*.

As noted in the manuscript, previous published experiments from this group had demonstrated the ability to rescue BRE function in the *osk* transcript in *trans* (Reveal et al., 2010). These experiments aim to provide additional insights into this phenomenon by studying rescue in *trans* of *osk* mutations other than BREs. The authors propose a broader implication for the functional effects of so-called "regulation in *trans*" beyond the BREs and the *osk* mRNA, albeit with a limited number of examples. The experiments also provide a convincing mechanism for the long-range regulatory interactions of these mRNAs – through PTB binding pyrimidine tracts (PTs). However, other mechanisms acting on *osk, grk*, and any other transcripts have not been ruled out.

Furthermore, the authors suggest that regulation in *trans* may be widespread, such that regulated assembly of different mRNAs into RNPs may make them amenable to regulation by factors that may not bind them directly. This would add a layer of complexity to our current understanding of post-transcriptional regulation, making the regulatory space accessible by such mRNAs much larger, and even more combinatorial in nature than previously thought. However, more evidence would be required to make this generalized conclusion.

Most of the results presented support the conclusions drawn by the authors, but a few supporting experiments are lacking:

Although the data presented in Figure 3 and Figure 4 are consistent with regulation in *trans* being facilitated by PTB-dependent RNP assembly, the authors have not explicitly demonstrated that this is mediated by direct binding of PTB to *oskar* mRNA, and that PTB binding is abolished in *osk* 3'PT mutants. Furthermore, claims that these mutations disrupt particle assembly should be provided in a more direct manner to support the authors' model.

IP of PTB followed by RTqPCR for WT *oskar* mRNA and the various 3'PT mutants of *oskar* mRNA would determine whether the candidate binding sites are functional, and whether PTB binding is abolished in the *osk*3'PT mutants.

If regulation in *trans* is PTB-dependent, the extent to which PTB binding is disrupted (by IP-RTqPCR or similar approaches) in the various *osk*3'PT mutants should correlate with their abilities to act as donors.

The role for Bru-mediated *osk* mRNA oligomerization elements in regulation in *trans* is proposed but not tested by the authors, citing that the Bru sites must be present to rescue their recipient mRNA (subsection “*osk* RNP interactions are required for rescue in trans”, second paragraph). However, earlier experiments in this manuscript have shown that rescue in *trans* is not limited to the BREs, so that other mRNA constructs are available to test this. For example, the rescue of *osk3'1004-1008* (Figure 1) by different donors with or without oligomerization elements could be assayed in a similar manner as the *osk C BRE^-^* used in Figure 3.

While the mutation of PTs in the *osk* 3' UTR had a marked effect on Grk protein levels, the authors did not address whether the reciprocal is true. That is, does mutation of the predicted PTs in the *grk* transcript lead to an increase in wild-type Osk protein levels? If the proposed mechanism of community regulation for transcripts in close proximity is true, effects of regulation in *trans* should be observed in both directions: *osk* and *grk* in close proximity would cross-regulate.

---

## [Author Response]

Summary:

*Both reviews on the manuscript have been completed and there is general concordance on several points. First, both reviewers feel that the work is of broad interest in taking earlier observations by your group showing that BREs' can be rescued in trans by another mRNA transcript, and extending these observations to show that other osk mutations can also be rescued in a manner potentially dependent on the PTB protein (and its binding sites). And, while both reviewers are inclined to think that indeed this idea is correct (if not the complete story), both reviewers feel that additional experiments are needed to support this specific claim. Most importantly, the distribution of the mutant RNA needs better characterization in the presence and absence of PTB (i.e. in the presence of PTB binding site mutations and with diminished PTB levels). One of the reviewers proposes the use of FISH to help support the proposed mechanism – two color FISH should help correlate the observed regulatory effects with colocalization. The second reviewer proposes some IP style experiments to address a similar point (where the amount of RNAs recovered correlate with the predicted strengths of the PTB binding sites, etc.). There were other experiments proposed to ask whether the nature of the mRNA interaction was reciprocal, and whether co-regulation happens in both directions. The reviewers agree that this would be interesting, but perhaps beyond the scope of the present work. In light of the broad agreement of the reviewers, we would be happy to receive a revised manuscript should there be additional experiments provided to support the proposed model that PTB binding is critical for both colocalization and coregulation, demonstrated more directly through biochemistry/visualization, and not simply through genetic approaches.*

I will address these main points here, and then respond in the context of the individual reviews to more detailed issues raised by the reviewers.

To provide biochemical evidence in support of the role for PTB in rescue/ regulation in *trans* we tried two approaches. The first was that suggested by reviewer 2, doing IPs of PTB to assess binding to the various mRNAs. Unfortunately, using the only anti-*Drosophila* PTB antibodies of which we are aware (Besse et al. Genes Dev 23, 195-207), we were unable to IP PTB. In the second approach, suggested by reviewer 1, we performed RNA binding assays with recombinant *Drosophila* PTB. In the revised Figure 3 we show that binding of PTB to the complete *osk* 3’ UTR is substantially reduced by the full set of PT mutations. In the revised Figure 4, we include two types of UV crosslinking binding assays. In one set of assays we used short segments of the *osk* 3’ UTR to ask if the PT sites within each segment mediated PTB binding. Using these short RNAs in the binding assay allows us to focus primarily on the PT sites mutated, in the absence of residual PTB binding to the remainder of the 3’ UTR. The results demonstrate that each subset of PT mutations does indeed reduce PTB binding. The second type of assay was designed to ask how strongly each subset of PT mutations disrupts PTB binding to the entire *osk* 3’ UTR, similar to the situation in vivo with the transgenes bearing the same subsets of mutations. Because we expected weaker effects (many PTB sites remain intact in the context of the full 3’ UTR), we used competition binding assays. The results show that subsets of the PT mutations do disrupt PTB binding. The degree to which PTB binding is reduced by each subset of mutations parallels the degree to which rescue in *trans* is reduced, at least in the sense that the subset of mutations with the greatest effect on PTB binding has the greatest effect on rescue in *trans*. The other subsets of mutations have lesser effects on both PTB binding and rescue in *trans*, but the differences within this group are relatively small and not statistically significant, and it seems unrealistic to think that we would obtain compelling evidence of a perfect correlation within this group between small differences in overall PTB binding and in rescue in *trans*. I think these experiments address the need for biochemical evidence to support the genetic data, and I hope you agree.

The imaging experiments are less straightforward. It seems that there are two types of information that the reviewers may want. One is confirmation that the various mRNAs are in the appropriate positions within the egg chamber to allow interactions (concentrated in the oocyte for the *osk/grk* interaction, and localized to the posterior pole of the oocyte to allow the *osk/osk* interactions). The second is superresolution imaging to visualize discrete particles in which interacting mRNAs are predicted to co-reside, and which might be disrupted when PTB binding is reduced.

The former type of experiment is simple. We have included a new supplemental figure to show that the mutations introduced into *osk* mRNA to interfere with RNP assembly do not prevent posterior localization of the mRNA, and so the *osk* mRNAs under study are colocalized. Published work from the Ephrussi lab (Besse et al. Genes Dev 23, 195-207) shows that *osk* mRNA localization is somewhat delayed but otherwise completely normal in egg chambers homozygous mutant for *heph*, the gene that encodes PTB. Because this is a more extreme situation than mutants heterozygous for *heph*, we cite their work but do not include experiments to monitor *osk* mRNA distribution in the *heph* heterozygotes. As for overlap in the distribution of *osk* mRNA and the GFP reporter mRNAs with *osk* 3’ UTR segments, multiple studies have shown that such reporters with the complete *osk* 3’ UTR are localized to the posterior pole of the oocyte just like *osk* itself (e.g. Besse et al. Genes Dev 23, 195-207). There is no strong posterior localization for the GFP reporters with only parts of the 3’ UTR, but the level of rescue in *trans* is extremely low for these transgenes, being detectable only in the indirect embryonic patterning assays. Thus, a low level of overlap in mRNA distributions could explain the observed rescue. Figure 5 shows that *osk* and *grk* are both highly enriched in early stage oocytes, with *osk* filling the entire oocyte. Thus, *osk* and *grk* have the potential to interact.

The superresolution experiments are not so simple. These appear to be what

reviewer 1 would like us to include, modeled on work recently reported from the Lehmann lab (Trcek et al., Nat Commun 6, 7962) in which the relative distributions of mRNAs localized to germ granules in embryonic pole cells were monitored. In particular, the reviewer suggests that we do 2 color FISH to test for colocalization of donor and recipient mRNAs, with the expectation that there will be homotypic clustering of mRNAs in discrete particles and that donor/recipient combinations that do not support rescue in *trans* will not show colocalization. To respond to these comments, it is useful to first explain what was done by Trcek et al. in their paper.

Trcek et al. did 2 color FISH, but the results did not demonstrate homotypic clustering of the germ granule mRNAs. Instead, the simultaneous detection of two mRNAs revealed extensive colocalization for most pairwise combinations. This colocalization is documented in Trcek et al. Figure 3, with a further example in Figure 4. The evidence for homotypic clustering comes from modeling, using data from different imaging experiments in which VasaGFP (protein) and single mRNAs were detected (not 2 color FISH). The modeling provides a prediction of the average distribution of mRNAs based on distance from VasaGFP, and does not exclude colocalization of fractions of the individual mRNAs. The overall conclusion from the work, which is best discerned from the Results section of Trcek et al., is that certain localized mRNAs have a bias towards homotypic clustering, but it is only a bias and is not absolute. For example, Trcek et al. state that “*pgc* mRNAs preferred to co-organize with other *pgc* mRNAs rather than mix with *nos* or *gcl* mRNAs.” Although a reader might be left with the impression that Trcek et al. found that different species of localized mRNAs were segregated one from another by homotypic clustering, the data do not show such a strong partitioning effect.

In the context of our work, the only experimental result that would disprove the regulation in *trans* model would be clear evidence that a combination of donor/recipient mRNAs that DOES display rescue in DOES NOT have any overlap in mRNA distributions. Thus, even if we obtained results comparable to the modeling of Trcek et al. (i.e. predicting some degree of homotypic clustering but not excluding colocalization), this would not disprove the model. Despite this argument that such experiments are unlikely to be conclusive, we would do them if they were practical. However, this would be a substantial project on its own. First, for the 2 color FISH we would have to generate an additional set of transgenic flies, so that both donor and recipient *osk* mRNAs each have their own unique sequence tags for detection (with our current transgenes, only the recipient has the unique GFP tag, while the donor mRNAs would have to be detected with an *osk* probe which would also detect the recipient mRNA). More significantly, we would have to establish a complex imaging system and develop novel reagents. It is worth noting that the Trcek et al. work involved collaboration of the Lehmann lab with two imaging labs, highlighting the technical sophistication and challenges of their approach. Furthermore, we would not simply be reproducing their assay system, as there are many differences between the tissues (embryos vs egg chambers) and the types of particles to be examined (assembled germ granules in embryos vs. the more diffuse and less highly organized germ granule precursors in developing oocytes). Notably, the reagent used for the Trcek et al. modeling studies – VasaGFP – is not suitable for our work. Vasa protein is a component of germ granules, but it is recruited by Osk protein and thus requires Osk expression (and activation of *osk* mRNA translation) as a prerequisite for assembly. We would need to develop an alternate reagent that could be used as a reference point for defining spatial organization, and it isn’t clear what molecule would have the required properties. The best candidate would be Osk protein itself, but this isn’t a viable option given that our experiments involve effects on Osk protein expression. We can’t simply add an Osk::GFP marker, as getting the protein expressed at the right position would require that it have the normal osk regulatory elements, which would then influence the expression of the mRNAs we are studying.

Reviewer #1:

*This manuscript is a follow up of the previous 2010 Developmental Cell paper (Dual Functions of BREs in Translational Regulation), the authors found that mutation of osk 3' UTR Bruno Recognition Element (BRE) results in developmental defects (measured in the number of cuticles in embryos). Coexpression of a second, protein null osk transcript (osk^54^) is able to rescue the developmental defects.*

*The authors carried out more mutations in the donor and the recipient osk mRNAs to try to understand what factors/elements are important for the rescue in trans. The authors characterized developmental defects and used GFP reporters to monitor rescue. There are two new conclusions from the paper. (1) PTB dependent mRNP assembly is essential for rescue in trans. (2) The translation regulation can affect other RNAs in the same RNP. Both of these are consistent with their in trans regulation model. The authors also perform an experiment on the IMP binding element mutant, which is not very conclusive. The second conclusion is novel and shows a potential physiological role of trans regulation. However, there are a few elements worth further clarification, especially with the first conclusion.*

*1) By deletion of nt121-150, the mutant cannot be rescued (Figure 1). The authors conclude that the 5' UTR is not important for being-rescued in trans. Since the deletion not only disrupts the long form osk protein, but also the 5' UTR of short form osk RNA, the transcript (D121-150) itself might have problems for translation regulation. In order to make a strong statement that the 5' UTR is not important for rescue of a recipient mRNA, additional mutations or experiments will be needed.*

There is a misunderstanding here about the experiments with the *osk* transcript with mutations in the 5’ UTR. We have shown two things. First, we find that the regulatory defect of this mutant transcript cannot be rescued in *trans*. Second, we show that this mutant transcript can, nevertheless, act as a donor for rescue in *trans* of other regulatory defects. We did not reach the conclusion noted by the reviewer, namely, that the *osk* 5’ UTR is not important for the ability of the mRNA to provide rescue in *trans*.

Our experiments do not directly address a possible contribution of this part of the mRNA (for example, by providing PTB binding sites to promote RNP assembly). Although we do find that the *osk* 3’ UTR alone provides a substantial degree of rescue in *trans*, this does not exclude a contribution from the *osk* 5’ region.

*2) It looks like that the RNP assembly is crucial for trans rescue. The mechanism proposed is that the donor and the acceptor mRNA need to be packaged together into an mRNP for this to happen. The authors showed development defects with GFP reporter expression. However, many mutations might result in defects in RNA localization or RNA assembly. Therefore, the distribution of the mutant RNA needs better characterization, for example, by FISH to donor and recipient, not just by qPCR to measure the total amount.*

Our response to this issue is described above.

*3) The authors argue that PTB dependent RNP assembly is crucial for the trans rescue. It is likely that the RNP assembly is necessary for the rescue. There are many ways that RNP assembly could be disrupted by the mutants. The point is how to prove it is PTB dependent. The authors mutated a set of PTB binding sites and showed rescue defects. There are two points the authors might want to show. First, these mutations may result in no or decreased PTB binding. The second might be hard to prove: so many mutations might disrupt something else. The authors also used a heterozygous mutant PTB to reduce the PTB level (is that true?). But as the author notes, PTB is essential for many RNA functions and might have indirect effects on osk.*

The PTB binding experiments requested by the reviewer are described above and included in the revised Figure 3 and 4.

As the reviewer notes, proving that the PT mutations only disrupt PTB binding and have no other effects that might influence rescue in *trans* is extremely difficult. This is why we took two approaches to ask if reduced PTB binding was an underlying cause of reduced rescue in *trans*. The logic behind these approaches is discussed in the manuscript, and the results provide very strong support for the conclusion that PTB binding to *osk* mRNA is required for rescue in *trans*. Nevertheless, we cannot exclude the possibility that the PT mutations have an additional effect on rescue in *trans*, but this does not weaken the main conclusion about the importance of PTB.

The use of heterozygous *heph* mutants (*heph* encodes PTB) to reduce PTB activity avoids the complication of indirect effects due to loss of PTB activity. Importantly, the *heph* heterozygotes have no obvious defects on *osk* expression or embryonic body patterning, and do not affect levels of protein produced by an *osk::GFP* transgene with no regulatory mutations (Figure 4). Thus, indirect effects seem extremely unlikely in this approach, even though they would be a very serious complication if we eliminated PTB activity altogether by using homozygous *heph* mutants.

*4) The inclusion of data from IBE mutations is puzzling and doesn't help the manuscript. It is idiosyncratic, as the authors suggest. I would remove it.*

We disagree with the reviewer on this point. The original IBE mutant is of considerable interest, given that it does not participate in rescue in *trans*. Knowing this information provides an opportunity to learn more about rescue in *trans*, and in particular to ask why this mutant has this property. Furthermore, the compelling evidence that the original IBE mutant is not a simple loss-of-function mutant provides important information for researchers studying mechanisms used to regulate translation of *osk* mRNA.

Reviewer #2:

*In this manuscript, Macdonald et al. have provided strong evidence in support of a model of post-transcriptional regulation in trans. Through their study of oskar and gurken mRNAs, the authors have characterized the regulatory elements required on donor and recipient mRNAs to allow for such regulation to take place. The authors go on to show that mutations in candidate PTB binding sites in the donor mRNA abolish the ability of the donor to regulate the recipient mRNA, and that regulation in trans is dependent on PTB dosage. The authors generalize their model by showing that oskar mRNA is capable of regulating another mRNA with which it co-localizes, gurken, in a manner that once again requires candidate PTB binding sites on the donor. They conclude that PTB-dependent particle assembly mediates regulation in trans.*

*As noted in the manuscript, previous published experiments from this group had demonstrated the ability to rescue BRE function in the osk transcript in trans (Reveal et al., 2010). These experiments aim to provide additional insights into this phenomenon by studying rescue in trans of osk mutations other than BREs. The authors propose a broader implication for the functional effects of so-called "regulation in trans" beyond the BREs and the osk mRNA, albeit with a limited number of examples. The experiments also provide a convincing mechanism for the long-range regulatory interactions of these mRNAs – through PTB binding pyrimidine tracts (PTs). However, other mechanisms acting on osk, grk, and any other transcripts have not been ruled out. Furthermore, the authors suggest that regulation in trans may be widespread, such that regulated assembly of different mRNAs into RNPs may make them amenable to regulation by factors that may not bind them directly. This would add a layer of complexity to our current understanding of post-transcriptional regulation, making the regulatory space accessible by such mRNAs much larger, and even more combinatorial in nature than previously thought. However, more evidence would be required to make this generalized conclusion. Most of the results presented support the conclusions drawn by the authors, but a few supporting experiments are lacking: Although the data presented in Figure 3 and Figure 4 are consistent with regulation in trans being facilitated by PTB-dependent RNP assembly, the authors have not explicitly demonstrated that this is mediated by direct binding of PTB to oskar mRNA, and that PTB binding is abolished in osk 3'PT mutants. Furthermore, claims that these mutations disrupt particle assembly should be provided in a more direct manner to support the authors' model. IP of PTB followed by RTqPCR for WT oskar mRNA and the various 3'PT mutants of oskar mRNA would determine whether the candidate binding sites are functional, and whether PTB binding is abolished in the osk3'PT mutants.*

*If regulation in trans is PTB-dependent, the extent to which PTB binding is disrupted (by IP-RTqPCR or similar approaches) in the various osk3'PT mutants should correlate with their abilities to act as donors.*

The biochemical assays to monitor effects of the PT mutations on PTB binding are described above.

*The role for Bru-mediated osk mRNA oligomerization elements in regulation in trans is proposed but not tested by the authors, citing that the Bru sites must be present to rescue their recipient mRNA (subsection “osk RNP interactions are required for rescue in trans”, second paragraph). However, earlier experiments in this manuscript have shown that rescue in trans is not limited to the BREs, so that other mRNA constructs are available to test this. For example, the rescue of osk 3'1004-1008 (Figure 1) by different donors with or without oligomerization elements could be assayed in a similar manner as the osk C BRE^-^ used in Figure 3.* We had hoped to take exactly this approach to test for a contribution of Bru binding sites to rescue in *trans*. Unfortunately, rescue in *trans* is very weak when both donor and recipient mRNAs have mutations in the cluster of regulatory sites near the 3’ end of the *osk* mRNA 3’ UTR. For example, co-expression of the *osk C BRE*^-^ mutant and the *osk3’1004-1008* mutant results in only very weak rescue in *trans*. This is perhaps not surprising, as the mutated sites are very close together and they may act in concert to assemble a regulatory complex.

We did test an *osk ABC BRE^-^* donor (with all BREs mutated to disrupt Bru-dependent *osk* mRNA oligomerization) for rescue of expression of the *osk3’1004-1008* mutant, and found no substantial rescue. Although this might appear to provide support for Bru-mediated oligomerization in rescue in *trans*, the fact that the C region BRE mutations have the same effect does not allow us to draw that conclusion.

*While the mutation of PTs in the osk 3' UTR had a marked effect on Grk protein levels, the authors did not address whether the reciprocal is true. That is, does mutation of the predicted PTs in the grk transcript lead to an increase in wild-type Osk protein levels? If the proposed mechanism of community regulation for transcripts in close proximity is true, effects of regulation in trans should be observed in both directions: osk and grk in close proximity would cross-regulate.*

Although this appears to be a logical extension to our work, the problem with the experiment is that *grk* mRNA is not strongly repressed. At early stages of oogenesis no Osk protein can be detected because repression is strong, while Grk protein is present (and is required) and so must be translated. There may be some degree of translational repression of *grk* mRNA (and Bru binds to *grk* mRNA, but much more weakly than Bru binds *osk* mRNA; Kim-Ha et al. Cell 81, 403-412; Filardo and Ephrussi, Mech Dev 120, 289-297), but it is clearly not the strong repression imposed on *osk* mRNA. Consistent with this, a GFP reporter mRNA with the *grk* 3’ UTR attached retains very strong GFP expression (our unpublished data), while adding either part of the *osk* 3’ UTR with BREs to the same GFP reporter largely eliminates GFP expression (Reveal et al. Dev Cell 18,496-502). Consequently, there is only a low level of direct repression of *grk* mRNA that has the potential to be exerted in *trans*, and this would be a very minor amount relative to the already strong repression of *osk* mRNA translation conferred in *cis* by the BREs.